# Silicon-based all-solid-state batteries operating free from external pressure

Zhiyong Zhang [1,9], Xiuli Zhang[2,9], Yan Liu[3,9], Chaofei Lan[1,9], Xiang Han [4] ✉,
Shanpeng Pei[1,5], Linshan Luo[1], Pengfei Su[1], Ziqi Zhang [6], Jingjing Liu[7],
Zhengliang Gong[8], Cheng Li[8], Guangyang Lin[1], Cheng Li[1], Wei Huang[1],
Ming-Sheng Wang [1,2] ✉ & Songyan Chen [1] ✉

Silicon-based all-solid-state batteries offer high energy density and safety but face significant application challenges due to the requirement of high external pressure. In this study, a $Li_{21}Si_5/Si-Li_{21}Si_5$ double-layered anode is developed for all-solid-state batteries operating free from external pressure. Under the cold-pressed sintering of $Li_{21}Si_5$ alloys, the anode forms a top layer ($Li_{21}Si_5$ layer) with mixed ionic/electronic conduction and a bottom layer ($Si-Li_{21}Si_5$ layer) containing a three-dimensional continuous conductive network. The resultant uniform electric field at the anode|SSE interface eliminates the need for high external pressure and simultaneously enables a twofold enhancement of the lithium-ion flux at the anode interface. Such an efficient ionic/electronic transport system also facilitates the uniform release of cycling expansion stresses from the Si particles and stabilizes bulk-phase and interfacial structure of anode. Consequently, the $Li_{21}Si_5/Si-Li_{21}Si_5$ anode exhibited a critical current density of 10 mA cm$^{-2}$ at 45 °C with a capacity of 10 mAh cm$^{-2}$. And the $Li_{21}Si_5/Si-Li_{21}Si_5|Li_6PS_5Cl|Li_3InCl_6|LCO$ cell achieve an high initial Coulombic efficiency of (97 ± 0.7)% with areal capacity of 2.8 mAh cm$^{-2}$ at 0.25 mA cm$^{-2}$, as well as a low expansion rate of 14.5% after 1000 cycles at 2.5 mA cm$^{-2}$.

The development and application of solid-state electrolytes (SSEs) with high ionic conductivity, such as sulfides and chlorides, have led to remarkable performance improvements in all-solid-state batteries (ASSBs)[1–4]. In particular, lithium metal/silicon-based ASSBs promise specific energies of over 500 Wh kg$^{-1}$[5,6]. However, in order to counteract the volume expansion and interfacial weakening of the anode, a high external pressure must be applied to provide a uniform force field.

Several recent studies have reported that a suitable stack pressure (e.g., 70 MPa) can increase the critical current density (CCD) of the lithium metal anode and thus inhibit the formation of lithium dendrites[7–9]. For the silicon (Si) anode, as shown in Supplementary Fig. 1a, a higher stack pressure (e.g., 370 MPa) is required to inhibit the volume expansion of the Si anode and stabilize the interface between the anode and the SSE[10–13]. The stack pressure mentioned above can provide an external pressure of tens of megapascals for the ASSBs

[1]Department of Physics, Collaborative Innovation Center for Optoelectronic Semiconductors and Efficient Devices, Key Laboratory of Low Dimensional Condensed Matter Physics (Department of Education of Fujian Province), Jiujiang Research Institute, Xiamen University, Xiamen, China. [2]State Key Lab of Physical Chemistry of Solid Surfaces, College of Materials, Xiamen University, Xiamen, China. [3]School of Semiconductor Science and Technology, South China Normal University, Foshan, China. [4]College of Materials Science and Engineering, Co-Innovation Center of Efficient Processing and Utilization of Forest Resources, Nanjing Forestry University, Nanjing, China. [5]Shandong Electric Power Engineering Consulting Institute Corporation, Jinan, China. [6]Science and Technology on Analog Integrated Circuit Laboratory, Chongqing, China. [7]Microsoft Corporation, One Microsoft Way, Redmond, WA, USA. [8]College of Energy, Xiamen University, Xiamen, China. [9]These authors contributed equally: Zhiyong Zhang, Xiuli Zhang, Yan Liu, Chaofei Lan. ✉e-mail: hanxiang@njfu.edu.cn; mswang@xmu.edu.cn; sychen@xmu.edu.cn

during cycling, which ensures its high performance[14,15]. In fact, maintaining high external pressure on the practical ASSBs is technically difficult and costly, which greatly hinders the scale-up process of ASSBs[1,16,17].

During the cycling of lithium metal ASSBs free from external pressure (Supplementary Fig. 1b), the lithium deposition behavior at the anode side is highly dependent on the electric field distribution at the interface[18,19]. Weakly-coupled interfaces are susceptible to voltage polarization, which induces inhomogeneous deposition of lithium metal and generate lithium dendrites, resulting in low CCD and insufficient capacity[20]. Different from the lithium plating in lithium metal anodes, Si anodes store lithium ion within the bulk phase of Si through an alloying process. This enables a significant higher CCD of Si anodes than lithium metal anodes[21,22]. However, during the first cycle, Si anodes can expand up to 300% in volume, leading to structural fragmentation of the anode[23,24]. This causes severe damage to the interface between the anode and the electrolyte, resulting in uneven distribution of lithium-ion flux and the formation of lithium dendrites (Fig. 1a). This challenge raises a critical question as to how to design an all-solid-state Si anode that is structurally and interfacially stable during cycling.

In this study, we present an anode design to homogenize the electric field of Si-ASSBs via $Li_{21}Si_5$ alloys, which eliminates the need for the force field from high external pressure and simultaneously enables a twofold enhancement of the lithium-ion flux at the anode interface. A $Li_{21}Si_5/Si-Li_{21}Si_5$ anode was proposed for ASSBs operating free from external pressure. As shown in Fig. 1b, the $Si-Li_{21}Si_5$ anode formed by mixing $Li_{21}Si_5$ alloy with Si at weigh ratio of 1:1 showed excellent three-dimensional continuous conductive network enabled by cold-pressed sintering. Compared to the pure Si anode, the $Li_{21}Si_5$ particle accelerates the self-discharge effect with Si particle, achieving a more uniform electrochemical sintering. However, this anode architecture still suffers regional cracking in bulk and lithium dendrite growth at the interface. Therefore, as shown in Fig. 1c, the $Li_{21}Si_5$ layer is pressed upon $Si-Li_{21}Si_5$ anode after cold-pressed sintering. It serves as a mixed ionic/electronic conductor layer, which homogenizes the surface electric field of the anode, facilitating uniform and rapid transportation of lithium ions to the surface of the $Si-Li_{21}Si_5$. The lithium ions then uniformly alloy with Si particles to form $Li_xSi$ (x<3.75) through the $Li_{21}Si_5$ alloy network within the electrode. The stress caused by the cycling expansion during the alloying process is uniformly released in the $Si-Li_{21}Si_5$ layer without damaging the interface. This ensures the stability of anode bulk and anode|SSE interface. As a result, the $Li_{21}Si_5/Si-Li_{21}Si_5|Li_6PS_5Cl|Li_3InCl_6|LCO$ cell delivered an high initial Coulombic efficiency (ICE) of (97 ± 0.7)% with areal capacity of 2.8 mAh cm$^{-2}$ at 0.25 mA cm$^{-2}$ and 45 °C, as well as a capacity retention reaching 80% after 183 cycles and 54.9% after 1000 cycles at 2.5 mA cm$^{-2}$.

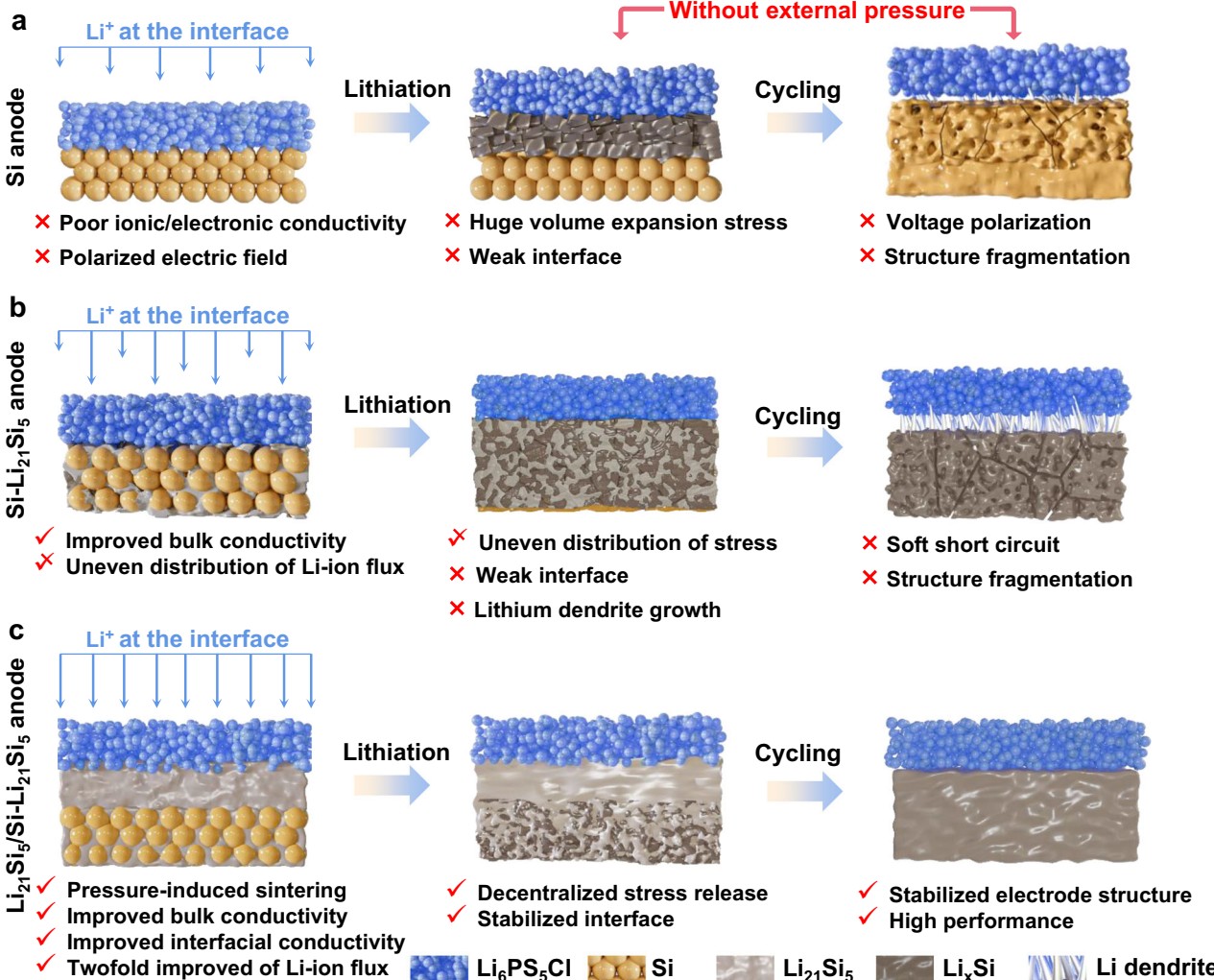

**Fig. 1 | Three different Si-anode designs and the advantages of pressure-free $Li_{21}Si_5/Si-Li_{21}Si_5$-ASSBs.** Schematic of the anode|SSE structures for Si-ASSBs (**a**), $Si-Li_{21}Si_5$-ASSBs (**b**), and $Li_{21}Si_5/Si-Li_{21}Si_5$-ASSBs (**c**).

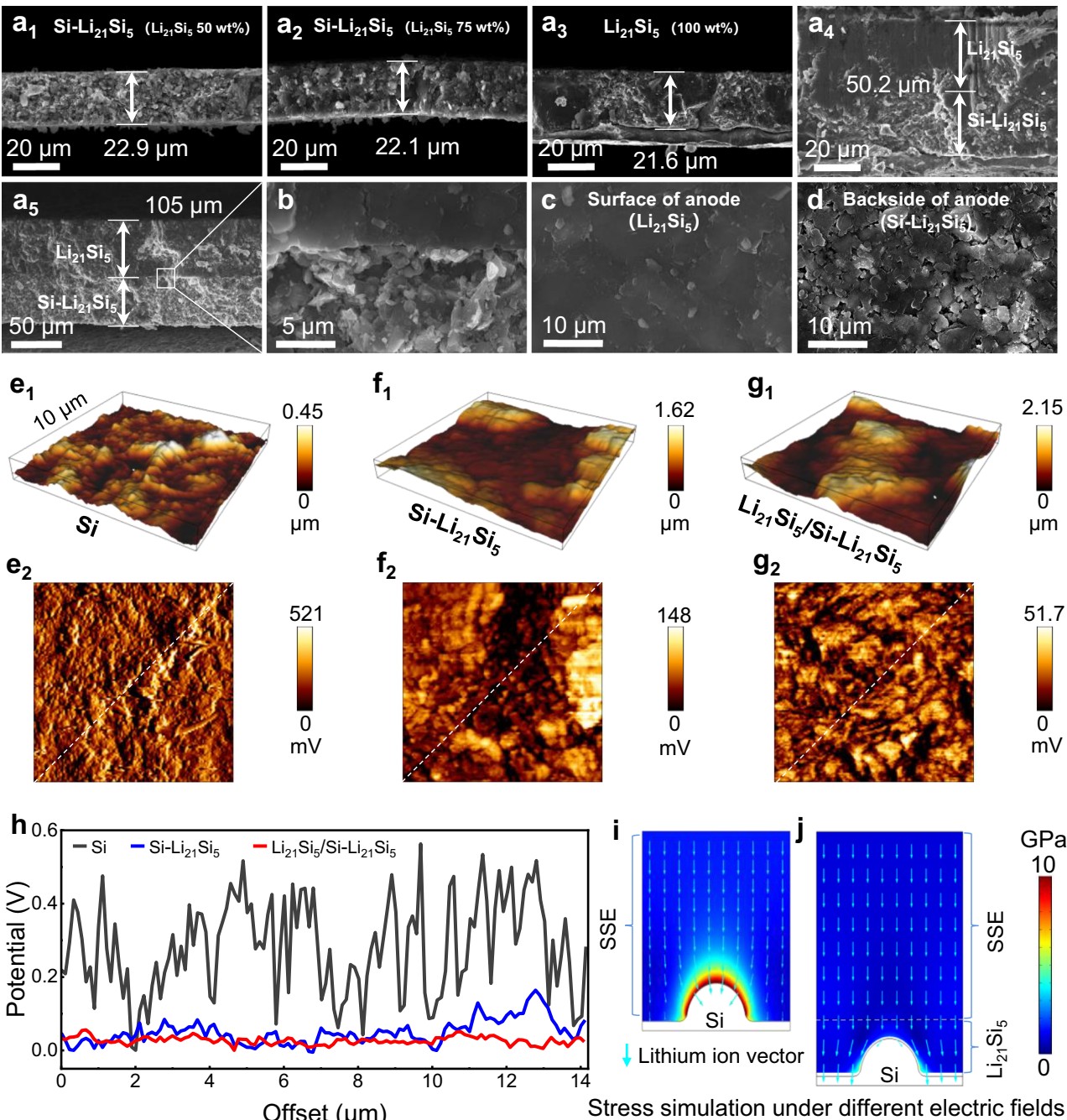

**Fig. 2 | Structural characterization and surface potential of the $Li_{21}Si_5$/Si-$Li_{21}Si_5$ anode.** Cross-section SEM images of Si-$Li_{21}Si_5$ electrodes with 50 wt% ($a_1$), 75 wt% ($a_2$) and 100 wt% ($a_3$) of $Li_{21}Si_5$. Cross-section SEM images of $Li_{21}Si_5$/Si-$Li_{21}Si_5$ electrode with thickness of 50.2 μm ($a_4$) and 105 μm ($a_5$). Enlarged cross-section (**b**), surface (**c**) and backside (**d**) SEM images of $Li_{21}Si_5$/Si-$Li_{21}Si_5$ anode. AFM images of Si anode ($e_1$), Si–$Li_{21}Si_5$ anode ($f_1$), and $Li_{21}Si_5$/Si–$Li_{21}Si_5$ anode ($g_1$), respectively. Area potential profile of Si anode ($e_2$), Si–$Li_{21}Si_5$ anode ($f_2$), and $Li_{21}Si_5$/Si–$Li_{21}Si_5$ anode ($g_2$), respectively. (**h**) line potential profiles in ($e_2$), ($f_2$), and ($g_2$). Stress and current distribution under Si|SSE model (**i**) and Si|$Li_{21}Si_5$|SSE model (**j**).

## Results and discussion

### Design and characterization of the $Li_{21}Si_5$/Si–$Li_{21}Si_5$ anode for ASSBs

Based on our previous work[22], the $Li_{21}Si_5$ powder was prepared by a spontaneous Li–Si alloying method, followed by its mixing with Si particles by stack pressure (50 wt% of $Li_{21}Si_5$ alloy was determined to be the optimal ratio for the anode). In this study, as shown in Fig. 2$a_{1-a3}$, 5 mg Si–$Li_{21}Si_5$ powders with 50 wt% ($a_1$), 75 wt% ($a_2$) and 100 wt% ($a_3$) of $Li_{21}Si_5$ was respectively weighed into the mould of ASSBs and cold-pressed under 600 MPa for 3 min to obtain electrode, their thicknesses were 22.9, 22.1 and 21.6 μm, respectively. As the $Li_{21}Si_5$ alloy content

increased, the electrodes became denser higher. And then, $Li_{21}Si_5$ electrode was pressed on the Si–$Li_{21}Si_5$ electrode (50 wt% of $Li_{21}Si_5$) under a stack pressure of 600 MPa, thus forming a $Li_{21}Si_5$/Si–$Li_{21}Si_5$ anode with the thickness of 50.2 μm (Fig. 2$a_4$). Further increasing the powder mass to 10 mg enabled the fabrication of electrodes with a thickness of 105 μm, in which the Si–$Li_{21}Si_5$ and $Li_{21}Si_5$ layers are 47.5 μm and 57.5 μm thick, respectivley. In Fig. 2$a_2$, $a_4$, and $a_5$, the theoretical maximum areal capacity of the $Li_{21}Si_5$/Si–$Li_{21}Si_5$ anode is 4.67, 9.37, and 18.75 mAh cm$^{-2}$, respectively. Surprisingly, the $Li_{21}Si_5$ particles in the anode were sintered together under the cold pressing (Fig. 2c, d). The surface scanning electron microscope (SEM) image of

Si–Li$_{21}$Si$_5$ electrode with 50 wt% Li$_{21}$Si$_5$ is shown in the Fig. 2d, the Li$_{21}$Si$_5$ particles (black) were distributed in a uniform manner around the Si particles (white), forming an excellent three-dimensional continuous conductive network enabled by cold-pressed sintering. Before applying stack pressure, the Li$_{21}$Si$_5$ particles (Supplementary Fig. 2a) and Si–Li$_{21}$Si$_5$ particles (Supplementary Fig. 2b) were uniformly dispersed. At this stage, the Si–Li$_{21}$Si$_5$ particles were unable to form close physical contact with each other. Supplementary Fig. 3a–g and Supplementary Fig. 3h demonstrate the sintering phenomenon and square resistance of Li$_{21}$Si$_5$ powder under different stack pressures. As the stack pressure was increased from 100 to 700 MPa, the neighboring Li$_{21}$Si$_5$ particles slowly fused together and reached the most suitable dense state at 600 MPa. At the same time, the value of square resistance dropped to a minimum of 0.05 Ω cm$^{-2}$. Supplementary Fig. 4 shows that Li$_{21}$Si$_5$ amorphization was more pronounced on the Si–Li$_{21}$Si$_5$ layer, as revealed by X-ray diffraction (XRD) patterns, compared to previous studies[22]. This indicates the promoted self-discharge behavior between Li$_{21}$Si$_5$ and Si particles. As shown in Supplementary Fig. 5, when Si–Li$_{21}$Si$_5$ is cold-sintered for a long time and then assembled into a ASSB with lithium metal, lithium ions in the Li$_{21}$Si$_5$ alloy continue to enter the bulk phase of Si to form Li$_x$Si. The voltage at the anode decreases continuously as the value of "x" increases. Compared to the case of 370 MPa, a greater stack pressure of 600 MPa is more conducive to the formation of a Li-Si conductive coating layer on the surface of Si particles. The reduction in Young's modulus of the Li$_{21}$Si$_5$ alloy is attributed to its saturated lithium content[25–27]. To better understand the self-discharge behavior, as shown in Supplementary Fig. 6, $^7$Li solid-state nuclear magnetic resonance (SSNMR) was used to probe the chemical state of Li$_{21}$Si$_5$ powder and Si–Li$_{21}$Si$_5$ powder. It is sensitive to both crystalline and amorphous phases. Based on the literature already reported by Key et al.[28], the homogeneous distribution of Si ions in the Li matrix gives rise to the most shielded Li environment(s), resulting in a broad resonance. The curve showed broad signals centered at 92.5, 9.5, and 5.2 ppm, the 92.5 ppm indicates the fully lithiated Si in the form of Li$_{21}$Si$_5$, and the 9.5 ppm and 5.2 ppm can be assigned to phases of Li$_{15}$Si$_4$ and Li$_{13}$Si$_4$, respectively. It indicates the presence of Li$_x$Si heterophase within the Li$_{21}$Si$_5$ phase. After cold-pressed sintering, the Si–Li$_{21}$Si$_5$ displayed a significantly enhanced signal at 9.5 ppm, suggesting that the Li-Si alloy is undergoing a transition from a high lithium state to a low lithium state. As a result, we demonstrate that the cold-pressed sintering and self-discharge effects facilitate the formation of integrated electrodes with three-dimensional ionic/electronic dual-conductor Li$_{21}$Si$_5$ layer and conductive networks in Si–Li$_{21}$Si$_5$ layer.

In order to further investigate the impact of the conductive network/layer, the Kelvin probe force microscopy (KPFM) was conducted, as illustrated in Fig. 2e–g. The roughness increases from 0.45 to 2.15 μm in Fig. 2e1-g1, which is likely attributable to the markedly lower Young's modulus of Li$_{21}$Si$_5$ alloys in comparison to that of Si. This property renders them highly malleable, facilitating their retention on the mold surface throughout the demolding process. The surface potential (Fig. 2e2-g2) and the line profiles (Fig. 2h) indicate that Si–Li$_{21}$Si$_5$ layer has a much more uniform potential distribution with a narrower range (0–148 mV) than pure Si (0–521 mV). However, as shown in Fig. 2d, h, pure Si particles are still present in the Si–Li$_{21}$Si$_5$ layer, leading to sharp fluctuations of its potential in local areas. Note that the uneven charge or potential distribution can result in the formation of "hot spots" that can induce the dendrite growth. In contrast, Li$_{21}$Si$_5$ layer exhibits the most uniform potential distribution with the narrowest range (0–51.7 mV), which can reduce the risk of dendrite growth and achieve uniform lithiation of Si particles.

In order to further analyze the distribution of current and stress, we simplified the Si–Li$_{21}$Si$_5$ anode as a Si anode, established two models (Si|SSE and Si|Li$_{21}$Si$_5$|SSE) via COMSOL, and calculated the current and stress distributions during the lithiation process based on

finite element simulations. As shown in Fig. 2i, j, the results demonstrate that during heterogeneous lithiation of the Si|SSE model, the Li$_x$Si (x<3.75) generated at the interface between the Si and SSE exhibits enhanced electrical conductivity, which attracts the lithium ions and causes a marked increase in expansion stress at the interfaces. In contrast, when the Si|Li$_{21}$Si$_5$|SSE model undergoes heterogeneous lithiation, the high conductivity of Li$_{21}$Si$_5$ ensures that the lithium ions are always in a homogenized electric field, resulting in a significantly smaller stress at the interface compared to the Si|SSE model.

As is well known, the lithiation process of pristine Si particles results in a huge volume expansion. Therefore, in order to achieve high performance of ASSBs operating without external pressure, it is critical to stabilize the structure of the Si anode during the first lithiation process. The ASSBs were assembled using SSE (Li$_6$PS$_5$Cl, Supplementary Fig. 7a) at the anode interface, lithium indium chloride SSE (Li$_3$InCl$_6$, Supplementary Fig. 7b) at the cathode interface, and lithium cobaltate (LiCoO$_2$, LCO) as the cathode material. To analyze the evolution of the Si anode during lithiation in the absence of external pressure, pure Si electrode, Si–Li$_{21}$Si$_5$ electrode, and Li$_{21}$Si$_5$/Si-Li$_{21}$Si$_5$ electrode were used as anodes, respectively. It should be noted that the Li$_{21}$Si$_5$/Si–Li$_{21}$Si$_5$ anode has a clearer bilayer structure at a high N/P ratio. In order to show the comparative performance of different electrodes, the electrochemical performance tests of the Li$_{21}$Si$_5$/Si–Li$_{21}$Si$_5$ anode in the following section were carried out based on an N/P ratio of 6.7.

As demonstrated in Fig. 3a–c, the three anodes display distinct characteristics during lithiation in ASSB at 3 mAh: (1) During the lithiation process at an elevated capacity of 0.5, 1.5, and 3 mAh, the lithiated region expanded slowly in the pure Si electrode due to the lack of an effective ionic/electronic conducting network. (2) Li$_{21}$Si$_5$ provides an efficient bulk phase conduction system for Si–Li$_{21}$Si$_5$ electrode, which can rapidly transport lithium ions to the Si particles. However, the transport of lithium ions is highly dependent on the distribution of the Li$_{21}$Si$_5$ particles. Note that lithium ions tend to react preferentially with the Si in the Li$_{21}$Si$_5$-enriched regions. This reaction produces a localized stress concentration during the electrochemical sintering process, leading to the formation of local cracks inside the electrode. (3) The electrode structure of the Li$_{21}$Si$_5$/Si–Li$_{21}$Si$_5$ electrode remained stable during the electrochemical sintering process. The Li$_{21}$Si$_5$ layer presents an excellent ionic/electronic conductive interlayer, which can homogenize the electric field and lithium-ion flux into the Si particles of the Si–Li$_{21}$Si$_5$. This also contributes to a uniform dispersion of the expansion stress of the Si particles. Further, the lithium ions uniformly and rapidly lithiated through the efficient conductive network of Si–Li$_{21}$Si$_5$. It is worth noting that this reaction process always occurs at the Si–Li$_{21}$Si$_5$ layer, which effectively prevents the expansion stress from damaging the interface between the Li$_{21}$Si$_5$ layer and SSE layer.

Figure 3d, e and Supplementary Fig. 8 show the electrochemical impedance spectra (EIS) of the above ASSBs after the initial cycle. EIS data was described in detail with the corresponding equivalent circuit, which includes the resistance of ohmic contact (Rs), ion-transfer resistance at SEI (Rsei), charge transfer resistance at the electrode/electrolyte interface (Rct), and the Warburg impedance (Zw)[19]. The electrodes of pure Si, Si–Li$_{21}$Si$_5$ electrodes, and Li$_{21}$Si$_5$/Si–Li$_{21}$Si$_5$ electrodes correspond to Rs of 55.3, 35.6, and 12.9 Ω, Rsei of 29.1, 12.6, and 5.3 Ω, and Rct of 201.8, 74.1, and 13.3 Ω, respectively. Figure 3f and Supplementary Fig. 9 show the lithium diffusion coefficient of the above ASSBs calculated from PITT. The Li$^+$ diffusion coefficient of Li$_{21}$Si$_5$/Si–Li$_{21}$Si$_5$|SSE|LCO was 2.09 × 10$^{-6}$ cm$^2$ S$^{-1}$ at 3.9 V, and was higher than 1.27 × 10$^{-6}$ cm$^2$ S$^{-1}$ and 0.08 × 10$^{-6}$ cm$^2$ S$^{-1}$ for Si–Li$_{21}$Si$_5$|SSE|LCO and Si|SSE|LCO. Supplementary Table 1 is comparison of diffusion coefficient in this work and different anode in published literature. It shows that the lithiated anode will have higher diffusion coefficients, and the Li$_{21}$Si$_5$ alloy has greater diffusion coefficients than them. These

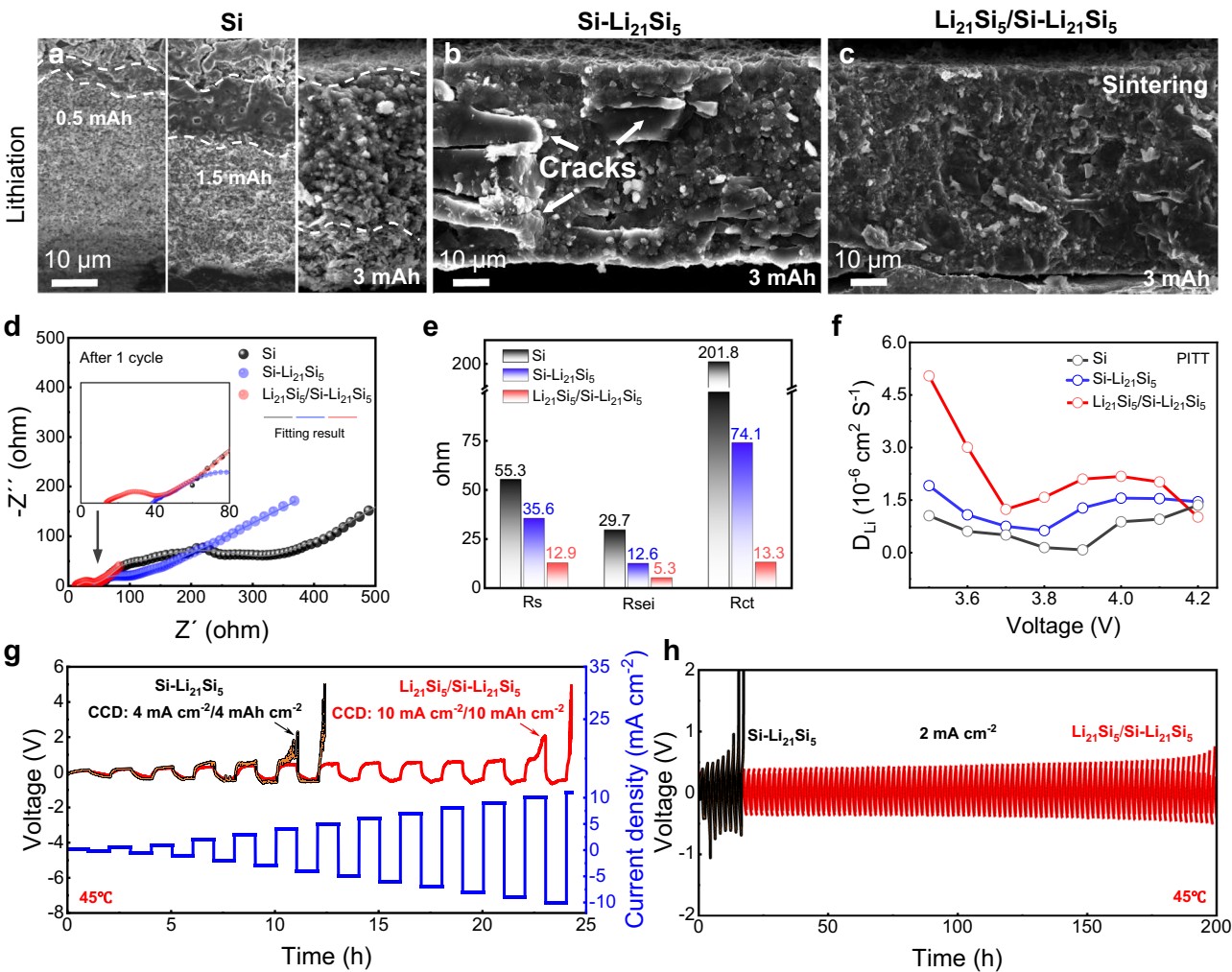

**Fig. 3 | Characterization of ionic/electronic transport property of the Li$_{21}$Si$_5$/Si−Li$_{21}$Si$_5$ anode for ASSBs.** Cross-section SEM images of Si (**a**), Si-Li$_{21}$Si$_5$ (**b**), and Li$_{21}$Si$_5$/Si-Li$_{21}$Si$_5$ (**c**) anode after lithiation of the anode|SSE|LCO cells. EIS test (**d**) and corresponding calculated resistance of the cells using Si, Si−Li$_{21}$Si$_5$, and Li$_{21}$Si$_5$/ Si-Li$_{21}$Si$_5$ anode after 1 cycle (**e**). **f** Lithium diffusion coefficient calculated from PITT. **g** CCD test for the cells of Si-Li$_{21}$Si$_5$|SSE|Li$_{21}$Si$_5$ and Li$_{21}$Si$_5$/Si−Li$_{21}$Si$_5$|SSE|Li$_{21}$Si$_5$. **h** Galvanostatic cycling of the above cells at 2 mA cm$^{-2}$.

results highlight the structural advantages of the Li$_{21}$Si$_5$/Si−Li$_{21}$Si$_5$ anode: Li$_{21}$Si$_5$ particles enhance the bulk-phase electric conductivity of the anode, while the Li$_{21}$Si$_5$ layer averages the electric field, disperses the expansion stress, and further stabilize the anode bulk and interface structure.

Figure 3g displays the charge-discharge profiles of the cells of Si−Li$_{21}$Si$_5$|SSE|Li$_{21}$Si$_5$ and Li$_{21}$Si$_5$/Si−Li$_{21}$Si$_5$|SSE|Li$_{21}$Si$_5$. At the end of the charge of the CCD test, a sharp voltage spike occurs, which can be ascribed to the expansion behavior of Si particles during lithiation. A CCD of 10 mA cm$^{-2}$ at a capacity of 10 mAh cm$^{-2}$ was achieved for Li$_{21}$Si$_5$/Si−Li$_{21}$Si$_5$ electrode, which was significantly larger than 4 mA cm$^{-2}$ at a capacity of 4 mAh cm$^{-2}$ of Si-Li$_{21}$Si$_5$ electrode. Figure 3h demonstrates the cyclic performance of the two types of cells at a current density of 2 mA cm$^{-2}$ and a capacity of 2 mAh cm$^{-2}$. The Si-Li$_{21}$Si$_5$ electrode showed a rapid increase in polarization voltage during the first 10 cycles. In contrast, the Li$_{21}$Si$_5$/Si−Li$_{21}$Si$_5$ electrode achieved 100 stable cycles. Furthermore, Supplementary Fig. 10 displayed 50 stable cycles at a current density of 4 mA cm$^{-2}$ and a capacity of 4 mAh cm$^{-2}$.

## Expansion evolution of the Li$_{21}$Si$_5$/Si−Li$_{21}$Si$_5$

It is known that Si anode will suffer dramatic volume changes during cycling, causing irreversible damage to the interface between the

anode and the SSE, which in turn leads to the voltage polarization. Among them, micrometer-sized Si particles will be accompanied by particle fragmentation process. To reveal the expansion evolution of the Li$_{21}$Si$_5$/Si−Li$_{21}$Si$_5$ during lithiation, we performed in-situ Transmission Electron Microscopy (in-situ TEM) to monitor its structural evolution during the initial charge process (Fig. 4, Supplementary Movie 1, and Supplementary Movie 2). The nanobattery setup in TEM involved a lithium metal counter electrode, the naturally formed lithium oxide (Li$_2$O) as a solid electrolyte, and a single Si particle from Si anode or Li-Si@Si particle from Li$_{21}$Si$_5$/Si−Li$_{21}$Si$_5$ anode as the working electrode, as depicted in Fig. 4a$_1$-a$_5$, 4b$_1$-b$_5$.

In the actual low-rate lithiation process, both the Si anode and the Li-Si@Si anode have enough time to react. Therefore, the time of lithiation is not limited in the in-situ TEM test. Due to the low intrinsic conductivity of pure Si particles (3 × 10$^{-5}$ S cm$^{-1}$), its lithiation process is highly dependent on the electrical contact condition. After 130 s, the Si particles began to lithiate from its contact area with Li$_2$O/Li electrode. After 160 s, the outer part of the particles underwent lithiation, causing intense volume expansion. Finally, after 180 s, the particles fragmented along the direction of lithiation due to the uneven distribution of lithium-ion flux. In contrast, the Si particles from the Li$_{21}$Si$_5$/Si−Li$_{21}$Si$_5$ anode have good ionic/electronic transport capability, because a Li−Si alloy layer is formed on the surface of the Si particles under the effect

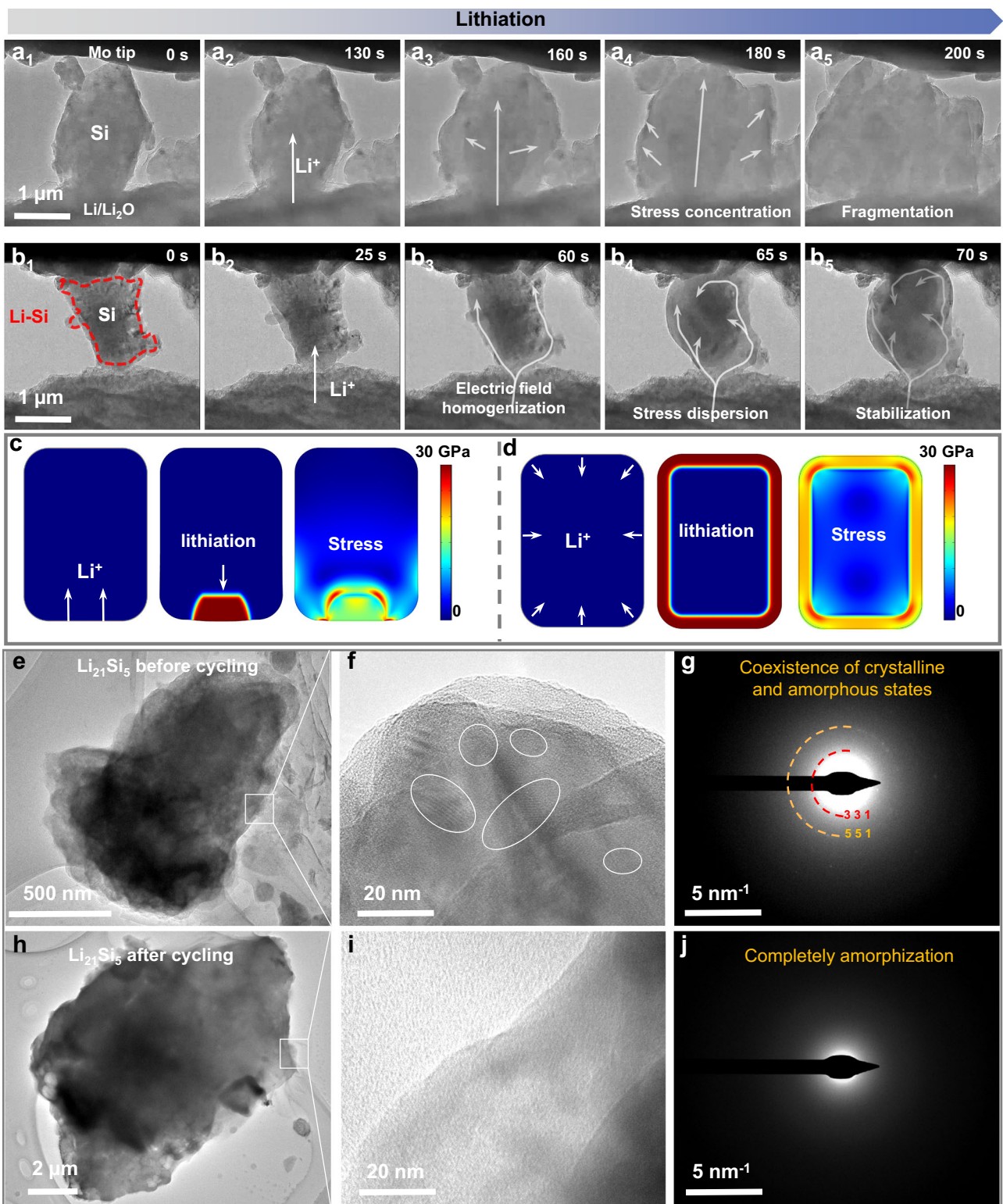

**Fig. 4 | Structure evolution of the $Li_{21}Si_5/Si$–$Li_{21}Si_5$ anode during cycling.** In-situ TEM observation of Si particle ($a_1$–$a_5$) and Li–Si@Si ($b_1$–$b_5$) particle during lithiation. Modeling and stress distribution of Si particle (**c**) and Li–Si@Si (**d**) particle during lithiation. HRTEM and selected area electron diffraction patterns of the $Li_{21}Si_5$ particle before **e**–**g** and after **h**, **j** cycling.

of self-discharge. As shown in the Supplementary Fig. 11, a 30–60 nm thick coating layer of $Li_xSi$ was observed on the surface of the Si particles after self-discharge. When the Li–Si@Si particle was brought into contact with $Li_2O$/Li electrode, the particle began to expand uniformly in the first 60 s, indicating rapid diffusion of lithium ions from the physical contact sites along the Li–Si coating layer to the Si core. After

65 s, the lithiation of Si was accelerated, forming more Li–Si alloy phase. After completion of lithiation, the Si particle expanded without fragmentation, and the lithiation rate was significantly faster. Benefiting from the pre-lithiation of the $Li_{21}Si_5/Si$–$Li_{21}Si_5$ anode during the cold-pressed sintering process, part of Li–Si@Si's volume expansion stress had been released in advance, while the subsequent uniform

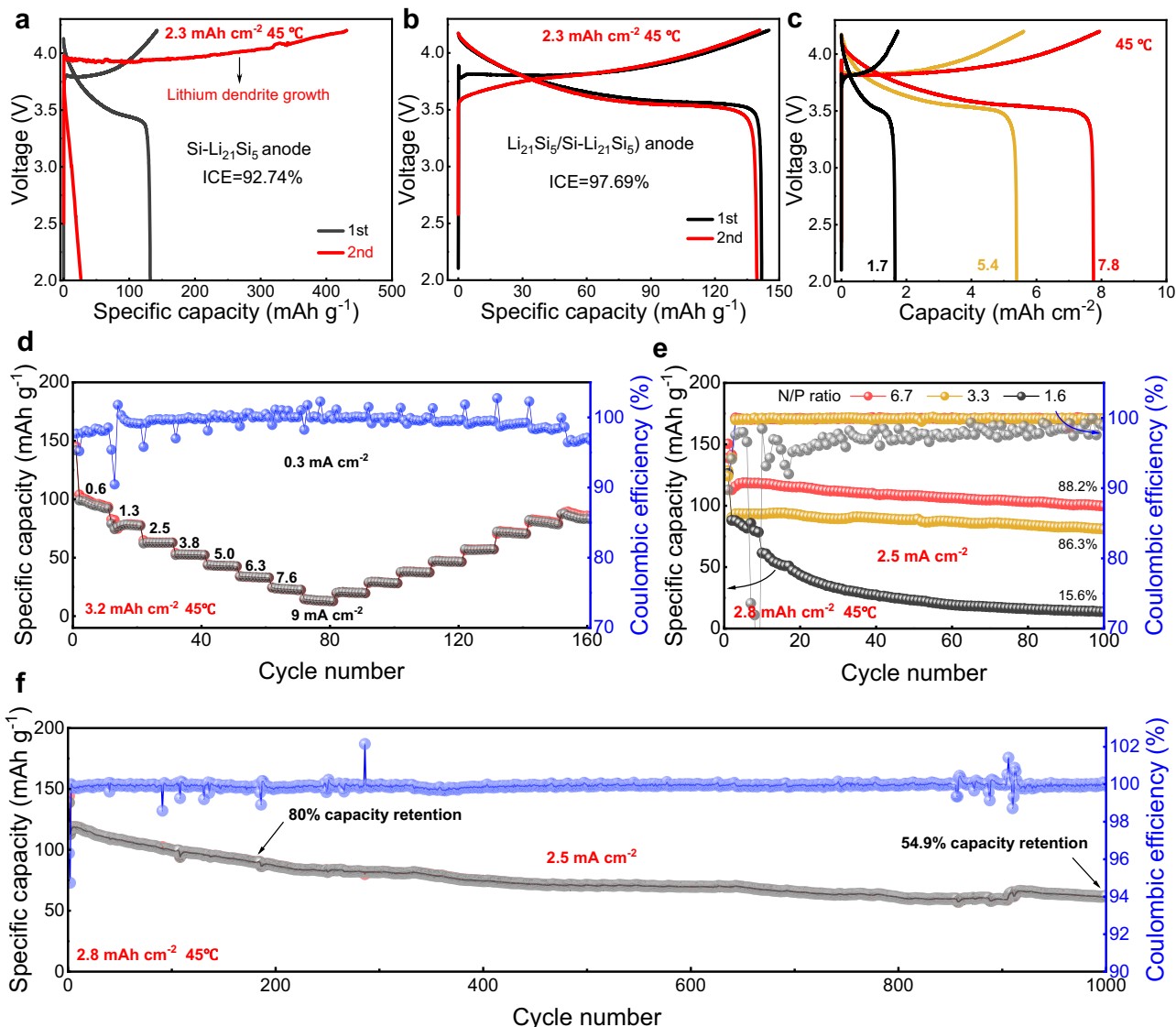

**Fig. 5 | Electrochemical properties of Li$_{21}$Si$_5$/Si-Li$_{21}$Si$_5$-ASSBs. a, b** Charge-discharge profiles of Si-Li$_{21}$Si$_5$-ASSB and Li$_{21}$Si$_5$/Si-Li$_{21}$Si$_5$-ASSB at 2.3 mA cm$^{-2}$ for cathode mass loadings of 16.8 mg cm$^{-2}$. **c**, Charge-discharge profile of Li$_{21}$Si$_5$/Si-Li$_{21}$Si$_5$-ASSB at different areal capacity. **d** Rate capability test. **e** Cycle performance of Li$_{21}$Si$_5$/Si-Li$_{21}$Si$_5$-ASSBs with different N/P ratios. **f** Long cycle performance of Li$_{21}$Si$_5$/Si-Li$_{21}$Si$_5$-ASSBs (N/P = 6.7). All of the above cells used Li$_6$PS$_5$Cl| Li$_3$InCl$_6$ as SSE and LCO as the cathode.

lithiation avoided fragmentation of the Si particles. As a result, the Li-Si@Si anode has a lower expansion rate. This suggests that the Li-Si layer on the surface of Li-Si@Si particles homogenizes the lithium-ion flux during the lithiation process, leading to efficient dispersion of expansion stresses.

To reveal the stress buildup/release mechanism of Si particle and Li-Si@Si particle, we performed modeling via COMSOL. As shown in Fig. 4c and Supplementary Fig. 12a, when the electrical contact of Si is limited, lithiation behavior occurs first in the contact region. The stress is accumulated at the interface between the outer Li$_x$Si and inner unreactive Si, reaching up to 30 GPa. Such a lithiation behavior likely leads to a buildup of stress, resulting in crack propagation. In contrast, as shown in Fig. 4d and Supplementary Fig. 12b, Li-Si@Si particle exhibits a distinct lithiation behavior. The expansion stress in the Si is uniformly distributed, with a maximum of only 20 GPa when the surface of the particles is evenly lithiated.

During the cycling process, Li$_{21}$Si$_5$ can also serve as a pre-lithiation particle for the ASSBs. Figure 4e-g shows discontinuous lattice fringes in the High-Resolution Transition Electron Microscopy (HRTEM)

images of Li$_{21}$Si$_5$ before cycling, corresponding to the diffraction rings of (331) and (551) obtained by selected area electron diffraction pattern. This indicates that the Li$_{21}$Si$_5$ before cycling was in a mixed state of amorphous and crystalline phases. After cycling, the HRTEM image did not show any obvious lattice fringes, indicating a complete amorphous state of the alloy (Fig. 4h-j). This conclusion is consistent with the amorphization of Si during the electrochemical sintering process. During the delithiation process, the potential of Li$_{21}$Si$_5$ is lower than that of Si and Li$_x$Si (x≤3.75) alloy, and consequently, Li$_{21}$Si$_5$ will lose lithium ions and eventually transformed into a Li$_x$Si alloy[29].

### Electrochemical properties and interface evolution of ASSBs operating free from external pressure

ICEs and cycling performance of Li$_{21}$Si$_5$/Si-Li$_{21}$Si$_5$-ASSBs with different LCO mass loadings were investigated operating free from external pressure. Figure 5a, b illustrates the ICEs and charge-discharge profiles of Si-Li$_{21}$Si$_5$-ASSB and Li$_{21}$Si$_5$/Si-Li$_{21}$Si$_5$-ASSB at 0.23 mA cm$^{-2}$ and a capacity of 2.3 mAh cm$^{-2}$. Their ICEs were 92.74% and 97.69%, respectively. However, the Si-Li$_{21}$Si$_5$-ASSB experienced a soft short circuit

during the second cycle, indicating that lithium dendrites were initiated at the interface. As shown in Supplementary Fig. 13a–c, the same phenomenon was observed in higher lithium content anode. In contrast, the $Li_{21}Si_5/Si$–$Li_{21}Si_5$-ASSB exhibits 99% CE in the second cycle, which can be ascribed to the homogenized lithium-ion distribution enabled by the ionic/electronic conducting $Li_{21}Si_5$ layer. The $Li_{21}Si_5/Si$–$Li_{21}Si_5$-ASSB in Fig. 5b demonstrate a marked overpotential during the initial charge cycle, which is likely attributable to side reactions between the $Li_{21}Si_5$ layer and the SSE, as well as concentration polarization between the $Li_{21}Si_5$ alloy and the cathode. The real-time pressure monitoring data of $Li_{21}Si_5/Si$–$Li_{21}Si_5$-ASSB was shown in Supplementary Fig. 14. After applying the minimum pressure load (0.8 MPa) of the equipment to limit the thickness of $Li_{21}Si_5/Si$–$Li_{21}Si_5$-ASSB, the pressure increased by 1.51 MPa in the first charge. In the next cycles, the rate of pressure change was slowly decreasing.

Homogeneous electric field at the interface and integrated bulk structure contribute to achieving excellent performance at high mass loading and high-rate. Figure 5c illustrates the charge-discharge profiles at 0.1 C for cathode mass loadings of 12 mg cm$^{-2}$, 36 mg cm$^{-2}$, and 54 mg cm$^{-2}$, respectively. At this time, their N/P ratios were 11, 3.5, and 2.4, with discharge capacities of 1.7 mAh cm$^{-2}$, 5.4 mAh cm$^{-2}$, and 7.8 mAh cm$^{-2}$, respectively. Figure 5d shows the rate performance of $Li_{21}Si_5/Si$–$Li_{21}Si_5$-ASSBs at different current densities. It delivered a specific capacity of 141.8 mAh g$^{-1}$ with ICE of 97.66% at current density of 0.3 mA cm$^{-2}$. When the current density was increased to 0.6 mA cm$^{-2}$, 1.3 mA cm$^{-2}$, 2.5 mA cm$^{-2}$, 3.8 mA cm$^{-2}$, 5.0 mA cm$^{-2}$, 6.3 mA cm$^{-2}$, 7.6 mA cm$^{-2}$, and 9 mA cm$^{-2}$, the discharge specific capacity was 99 mAh g$^{-1}$, 79 mAh g$^{-1}$, 62.2 mAh g$^{-1}$, 52.7 mAh g$^{-1}$, 43.3 mAh g$^{-1}$, 33.8 mAh g$^{-1}$, 24.3 mAh g$^{-1}$, and 14.5 mAh g$^{-1}$, respectively. The cycling performance is more stable at higher current density, indicating the excellent high-rate performance. Furthermore, Supplementary Fig. 15 illustrates the rate performance at high areal capacity with a cathode mass loading of 60 mg cm$^{-2}$. When the current density was 0.3 mA cm$^{-2}$, 0.6 mA cm$^{-2}$, 1.3 mA cm$^{-2}$, 2.5 mA cm$^{-2}$, 5.0 mA cm$^{-2}$, 7.6 mA cm$^{-2}$, and 10.2 mA cm$^{-2}$, the areal capacity was 7.8 mAh cm$^{-2}$, 6.7 mAh cm$^{-2}$, 5.7 mAh cm$^{-2}$, 3.5 mAh cm$^{-2}$, 1.2 mAh cm$^{-2}$, 0.3 mAh cm$^{-2}$, and 0.02 mAh cm$^{-2}$, respectively.

To analyze the effect of the lithiation depth of Si in the $Li_{21}Si_5/Si$–$Li_{21}Si_5$ anode on its cycling performance, as shown in Fig. 5e, ASSBs with N/P ratios of 1.6, 3.3, and 6.7 were tested at the same current density and cathode capacity mass loading. As the N/P ratio increased, the ICE of the cells exhibited values of 89.8%, 91.7%, and 96.3%, respectively. After 100 cycles, their capacity retentions were 15.6%, 86.3%, and 88.2%, respectively. These findings indicate that saturated lithiation of Si directly results in low ICE and capacity decay of the cells during cycling operating free from external pressure. Supplementary Fig. 16 displays the cross-section SEM images of above $Li_{21}Si_5/Si$–$Li_{21}Si_5$ anode after cycling with thicknesses of 37.4, 64.9, and 121.4 μm, corresponding to expansion rate of 69.2%, 29.3%, and 15.6%, respectively. It is evident that the electrode with N/P ratios of 1.6 exhibit pronounced cracking, which is likely the underlying cause of their performance degradation. Indeed, the majority of commercially available Si anode are designed to circumvent the phenomenon of saturated lithiation of Si.

Furthermore, the cycling performance of the $Li_{21}Si_5/Si$–$Li_{21}Si_5$-ASSB (N/P = 6.7) was tested with a cathode mass loading of 20 mg cm$^{-2}$, which delivered a discharge specific capacity of 138.9 mAh g$^{-1}$ (Fig. 5f). The capacity retentions were 80% (183 cycles) and 54.9% (1000 cycles) at 2.5 mA cm$^{-2}$, and average CEs were 99.86% and 99.92%, with standard deviations of 0.0048 and 0.018, respectively. In contrast, the ASSB made of pure Si as anode exhibited a specific capacity of 80.5 mAh g$^{-1}$ during the first discharge operating free from external pressure, but its ICE was only 66.9% (Supplementary Fig. 17a). Unfortunately, the cell failed abruptly after only 20 cycles. Moreover, the discharge specific capacity of the Si-$Li_{21}Si_5$-ASSB dropped from 73.2 mAh g$^{-1}$ to 49.5 mAh g$^{-1}$ after 20 cycles at a current density of 1.3 mA cm$^{-2}$. A soft short circuit occurred after 30 cycles, and the cell suddenly failed after 685 cycles (Supplementary Fig. 17b).

The EIS tests were performed on the Si-ASSB, Si-$Li_{21}Si_5$-ASSB, and $Li_{21}Si_5/Si$–$Li_{21}Si_5$-ASSB (Supplementary Fig. 18a). The results of the $Li_{21}Si_5/Si$–$Li_{21}Si_5$-ASSB before and after cycling showed an increase in Rs from 12.9 to 38.6 Ω, Rsei from 5.3 to 23.3 Ω and Rct from 13.3 to 38.9 Ω (Supplementary Fig. 18b). The slowly increasing impedance demonstrated that the anode structure and interface of the $Li_{21}Si_5/Si$–$Li_{21}Si_5$-ASSB remained stable. In contrast, the Rct of Si-ASSB and Si-$Li_{21}Si_5$-ASSB, especially the former, had exponentially increased, indicating severe damage to the anode structure and interface. It is worth noting that the Rs of the Si-$Li_{21}Si_5$-ASSB was much lower than the initial value of 36.5 Ω before cycling, suggesting a soft short circuit during cycling, which is consistent with the conclusion of Supplementary Fig. 17. Finally, Supplementary Fig. 19 and Supplementary Table 2 compare the cycling performance of $Li_{21}Si_5/Si$–$Li_{21}Si_5$-ASSB with the published literatures.

To further analyze the mechanism of $Li_{21}Si_5/Si$–$Li_{21}Si_5$ anode, the ASSBs in the Supplementary Fig. 18 was disassembled after cycling. Figure 6a–g displays the results of its X-ray photoelectron spectroscopy (XPS) and cross-section SEM characterizations. After cycling, (Li-)Si-$PS_4^{3-}$ (100.8 eV) and Li-Si (98.2 eV) were found on the anode surface (Fig. 6a), which was related to the electrochemical decomposition behavior of the sulfide electrolyte at the interface with the $Li_{21}Si_5$ alloy. This is further supported by the appearance of $Li_2S$ in the S 2p orbitals and $PO_4^{3-}$ in the P 2p orbitals, as shown in Fig. 6b, c. Due to the low Young's modulus and high ionic/electronic conductivity of the $Li_{21}Si_5$ layer, the $Li_6PS_5Cl$ maintains stable physical contact with the $Li_{21}Si_5$ layer. This results in significantly weaker decomposition behavior of the $Li_6PS_5Cl$ at the interface compared to previous reports[22], which is more conducive to lithium ions transport at the interface. Figure 6d–g and Supplementary Fig. 20 show the cross-section SEM images and elemental mapping of $Li_{21}Si_5/Si$–$Li_{21}Si_5$-ASSB after cycling operating free from external pressure. After cycling, the interface between the anode and the electrolyte remained in close contact without any visible cracks. The anode had a thickness of 120.3 μm and an expansion rate of 14.5%. In contrast, as shown in Fig. 6h, i, the Si-$Li_{21}Si_5$ ASSB showed significant lithium dendrite growth behavior at the anode|SSE interface after cycling. Even if a reactive region with good electric contact existed at the anode, the structure of the anode would be damaged by the large regional stresses, resulting in the formation of a large number of cracks (Supplementary Fig. 21). The fragmentation and volume change of the anode further weakens the interfacial contact between the anode and the SSE, which in turn generates a larger polarization voltage, leading to the formation of lithium dendrites due to lithium-ion concentration at the "hot spots".

In this study, we developed a $Li_{21}Si_5/Si$–$Li_{21}Si_5$ double-layered anode for ASSBs operating free from external pressure. The cold-pressed sintering of $Li_{21}Si_5$ powders at 600 MPa can facilitate the construction of a structurally stable $Li_{21}Si_5/Si$–$Li_{21}Si_5$ anode with a mixed ionic/electronic conductor layer (the $Li_{21}Si_5$ layer) and three-dimensional continuous conductive networks ($Li_{21}Si_5$ network in the Si-$Li_{21}Si_5$ layer). The $Li_{21}Si_5$ layer homogenizes the surface electric field of the anode, facilitating uniform and rapid transportation of lithium ions to the surface of the Si-$Li_{21}Si_5$ and avoiding the dendrites penetration into the SSE. Sufficient electrical contact will accelerate the self-discharge behavior between $Li_{21}Si_5$ and Si particles, and avoid stress concentration during cycling of Si particles, which stabilizes the structure of anode. It enabled the ASSBs to achieve an ICE of (97 ± 0.7)% at 0.25 mA cm$^{-2}$ and 45 °C. The error of ICE was estimated based on the maximum and minimum values. Stable cycles were achieved for first discharge capacities of 2.8 mAh cm$^{-2}$ with low expansion rate of 14.5%. The $Li_{21}Si_5/Si$–$Li_{21}Si_5$-ASSBs also achieved excellent performance at different rates. In

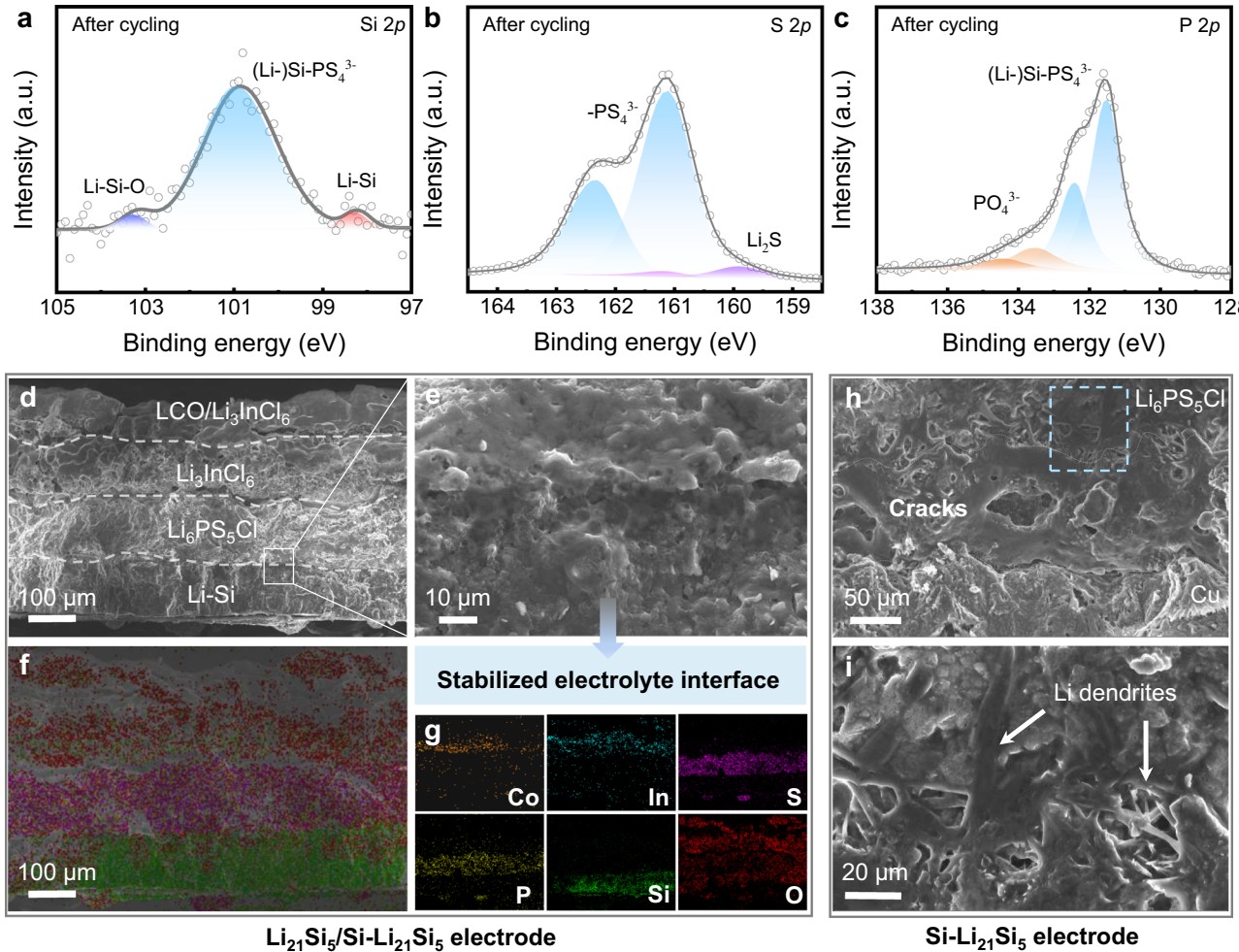

**Fig. 6 | Interface evolution of Li$_{21}$Si$_5$/Si-Li$_{21}$Si$_5$-ASSBs. a–c** XPS tests of Li$_{21}$Si$_5$/Si-Li$_{21}$Si$_5$-ASSBs after cycling. **d, e** and **f, g** Cross-section SEM images and EDS mappings of Li$_{21}$Si$_5$/Si-Li$_{21}$Si$_5$-ASSBs after cycling. **h** Cross-section SEM images of Si-Li$_{21}$Si$_5$-ASSBs after cycling. **i** Enlarged SEM image of the boxed area of Fig. 6h.

addition, through in-situ/ex situ testing, we found that the structural fragmentation of the Si anode will directly disrupt the interfacial electrical contact between the SSE and the anode during cycling, resulting in a larger polarization voltage, which in turn induces the growth of lithium dendrites.

## Methods

### Synthesis of materials and electrodes

The Si powder (1 μm, 99.9% Xuzhou Jiechuang), LCO (99.99%, Kejing), Li$_3$InCl$_6$ (200 nm, water-mediated synthesis[22]), Li$_6$PS$_5$Cl (500 nm, 99.9%, Mache Power) and ASSB mould (WuHan Chuangneng) were stored in an Ar-filled glove box with moisture and O$_2$ contents below 0.1 ppm. Preparation of electrodes and assembly /disassembly of ASSBs were carried out in Ar-filled glove box. Firstly, the Si powder and the lithium tablets (99.9%, Canrd) were weighted with a molar ratio of 5:21. Put the lithium tablets on a heating plate. Spread the Si powder (thickness is ~0.5 cm) on the edge of lithium ensuring physical contact between them. Heat the plate to the 400 °C and trigger the spontaneous reaction between Si powder and molten Li. As the reaction was completed, the Li$_{21}$Si$_5$ was obtained. Secondly, Li$_{21}$Si$_5$ and Si were weighed and mixed homogeneously according to the Li$_{21}$Si$_5$ alloy ratio of 0 wt%, 50 wt%, 75 wt%, and 100 wt%, respectively. 5 mg/10 mg of the above powders were respectively weighed into the mould of ASSBs and cold-pressed under 600 MPa for 3 min to obtain the pure Si electrode, Si–Li$_{21}$Si$_5$ electrode, and pure Li$_{21}$Si$_5$ electrode. And then, 5 mg/10 mg of the Li$_{21}$Si$_5$ alloy was placed on the surface of Si–Li$_{21}$Si$_5$

electrode, 600 MPa was applied and held for 3 min to obtain the Li$_{21}$Si$_5$/Si–Li$_{21}$Si$_5$ electrode.

### All-solid-state ASSBs assembling

30 mg of Li$_6$PS$_5$Cl was weighed and placed on the surface of the above electrodes and pre-pressed. 50 mg of Li$_3$InCl$_6$ was weighed, placed on the surface of Li$_6$PS$_5$Cl and pre-pressed. The moulds were cold-pressed at 370 MPa to obtain the bilayer SSE. LCO and Li$_3$InCl$_6$ were weighed at a mass ratio of 6:4, mixed and hand-milled for 10 min. The powders were then placed on the electrolyte surface to obtain cathode layers. Then the moulds were cold-pressed at 350 MPa and held for 3 min to obtain ASSBs. If Li$_6$PS$_5$Cl powder was used as SSE and Li$_{21}$Si$_5$ alloy as the cathode, the cells of Si–Li$_{21}$Si$_5$|SSE|Li$_{21}$Si$_5$ and Li$_{21}$Si$_5$/Si–Li$_{21}$Si$_5$|SSE|Li$_{21}$Si$_5$ was obtained.

### Characterization

The morphology and corresponding element distribution were characterized by field-emission scanning electron microscopy (FESEM, SIGMA-HD) and an energy-dispersive X-ray detector, respectively. All the HRTEM experiments were conducted on a FEI Talos F200s operating at 200 kV. AFM images and area potential profile were simultaneously collected using the Kelvin probe force microscopy (KPFM, Bruker NW4). The image scan rate was set at 0.3 s per line with a resolution of 128 × 128 pixels. The phase structure was detected by X-ray diffraction (XRD, Rigaku Ultima IV) with a Cu Kα source. $^7$Li SSNMR were conducted on a Bruker NEO-600WB NMR spectrometer.

Samples were packed in 1.3 mm rotors and spun at a speed of 15 kHz. The XPS was performed on the X-ray photoelectron spectrometer (XPS, Thermo Scientific K-Alpha).

### In-situ TEM tests

The in-situ TEM experiments were carried out using a FEI Talos F200s TEM equipped with a TEM-STM holder (ZepTools Co. Ltd, China), which was able to piezo-driven manipulation and electrical biasing. Si or Li−Si@Si particles were loaded on a Mo tip as the working electrode, while a Cu tip attached with Li metal was mounted on the other end of the holder serve as counter electrode. The samples were assembled inside Ar-filled glove box. Then the holder was kept in Ar-filled airtight box for transfer, and inserted into TEM within 5 s. During the insertion process, a thin $Li_2O$ layer formed on the surface of Li metal. When the Si or Li−Si@Si particle was brought into contact with $Li_2O$ layer which served as solid electrolyte, a solid-state nanobattery was in-situ assembled. A biasing voltage (+10 V) was applied on the Li metal to initiate the lithiation process of Si, by reversing the biasing voltage, the delithiation process of Si occurred.

### Electrochemical measurements

The electrochemical impedance of ASSBs was measured by the CHI660E electrochemical workstation from 100 kHz to 0.1 Hz with an amplitude of 10 mV. Galvanostatic cycling of symmetrical cells and ASSBs at different current densities were conducted on a LAND (CT2001A) battery tester. The charge/discharge cycling performance of ASSBs was carried out in 2.0–4.2 V (vs. Li/Li$^+$). The lithium diffusion coefficient was calculated from a LAND (CT2001A) battery tester using potentiostatic intermittent titration technique (PITT) with a potential step of 2 mV each step lasting 15 min. All of the above electrochemical tests were performed in a glove box at 45 °C with $H_2O$ and $O_2$ levels below 0.1 ppm.

### Modeling and simulation

The finite element analysis based on the commercial software COM-SOL Multiphysics was adopted by the method previously reported in the research[30]. In Fig. 2i, j, a periodicity simplified 2D structure was established to simulate the phase transition due to the chemical/electrochemical reaction caused by lithium ions and electron transport in materials.

The current density (derived from Li-ion transport) of this model is determined by Ohm's law:

$$\mathbf{J} = \sigma \nabla \boldsymbol{\varphi} \tag{1}$$

Where $\mathbf{J}$ is current density and $\boldsymbol{\varphi}$ is potential distribution of this model, $\sigma$ is the resistance which is related to electronic and ionic conductivity of material.

To simulate the phenomenon of phase interface transition associated with the process of charge transfer (reaction between lithium ions and Si), we used the simplified Navier−Stokes equation, called Level-Set, to describe the whole system as following expression shown:

$$\frac{\partial \phi}{\partial t} + \mathbf{u} \cdot \nabla \phi = \gamma \nabla \cdot [\varepsilon \nabla \phi - \phi(1-\phi)\frac{\nabla \phi}{|\nabla \phi|}] \tag{2}$$

Where $\gamma$ is a reinitialization parameter to assist equation solved easily, $\varepsilon$ is parameter controlling interface thickness to describe the thickness of phase interface, $\mathbf{u}$ is the velocity field of phase interface. $\phi$ is used to describe the phase state of material, and it is defined as:

$$\phi = \begin{cases} 0, Reacted\ Si \\ 1, Si \end{cases} \tag{3}$$

One should note that phase transition is controlled by velocity field $\mathbf{u}$, which is related to the current $\mathbf{J}$ in Si, and we have,

$$\mathbf{u} = \frac{\mathbf{J}M_{Li}}{F\rho} \tag{4}$$

Where $M_{Li}$ is the relative atomic mass of lithium, $F$ is the Faraday constant, $\rho$ is the mass participating in the Si reaction per unit volume. We then use the previous study to solve the stress-strain simulation of the modified system[30].

In Fig. 4c, d, an axisymmetric model was built from the perspective of the number of lithium ions transport sites on the surface of the Si particles, and the mechanical properties and expansion characteristics of the material were studied. The parameters involved in the model were interpolated according to the parameters of Li−Si alloy materials existing in the crystal database and literature, including density, Young's modulus, etc. The stress distribution model of whole model was built using COMSOL Multiphysics.

### Reporting summary

Further information on research design is available in the Nature Portfolio Reporting Summary linked to this article.

## Data availability

The authors declare that all the relevant data are available within the paper and its Supplementary Information file or from the corresponding author upon request. Source data are provided with this paper.

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

## Acknowledgements

The authors acknowledge the support from the Inital Energy Science &Technology (Xiamen). This research was supported by the National Natural Science Foundation of China (Grants No. 52172240, 22209075).

## Author contributions

Z.Z. and S.C. conceived the concept. Z.Z. carried out the synthesis and performed materials characterizations and electrochemical measurements. Z.Z., S.C., M.W., Y.L., X.Z., and C. Lan inspired the synthesis method. Z.Z., C. Lan, Y.L., S.P., P.S., L.L., G.L., C.Li, Z.Q.Z., and W.H. conducted SEM, XRD and XPS test. C. Lan and Z.Z. conducted COMSOL modeling. Z.Z., Z.G., and C.L. conducted real-time pressure monitoring test. X.Z., M.W., and Z.Z. conducted HRTEM and in-situ TEM. Z.Z., M.W., S.C., X.H., and J.L. co-wrote the paper. All authors participated in the analysis of the experimental results.

## Competing interests

The authors declare no competing interests.
