## [Transparent Peer Review file · Nature Communications]

Silicon-based all-solid-state batteries operating free from external pressure

Corresponding Author: Professor Songyan Chen

Version 0:

Reviewer comments:

Reviewer #1

(Remarks to the Author)

This work introduces a concept for operating a stable all-solid-state battery without external pressure by forming a $\text{Li}_{21}\text{Si}_5/\text{Si}-\text{Li}_{21}\text{Si}_5$ bilayer. Although the proposed bilayer showed improved battery performance compared to the anode performance of a silicon-only or Li_{21}Si single layer, the concept exhibits several technical limitations. As shown in the cross-sectional image in Figure 2, the thickness of the $\text{Li}_{21}\text{Si}_5$ used as a protective layer is $50\ \mu\text{m}$, which is not realistic. In fact, the thickness of the $\text{Si}-\text{Li}_{21}\text{Si}_5$ layer is also $50\ \mu\text{m}$, making the total anode layer thickness $100\ \mu\text{m}$. Considering the content of Si where the electrochemical reaction occurs, it is unclear what advantages can be gained, because the energy density in this cell configuration is expected to be very low. Additionally, there was no in-depth electrochemical property analysis about $\text{Li}_{21}\text{Si}_5$, such as its ionic conductivity and electrical conductivity. Furthermore it is difficult to find any advancements in this paper compared to the authors' recently published work (Energy Environ. Sci., 2024, 17, 1061). To be considered for publication, a clear explanation of the new advancements and an in-depth additional analysis are required. As it stands, I cannot support the publication of this paper in Nature Communications

Reviewer #2

(Remarks to the Author)

The stable operation of the Si anode at low pressure is important for the commercialization of all-solid-state cells. The following comments have to be addressed before paper acceptance.

1. What is the impact of $\text{Si}/\text{Li}_{21}\text{Si}_5$ weight ratio in $\text{Si}-\text{Li}_{21}\text{Si}_5$ on the electrochemical performance of $\text{Li}_{21}\text{Si}_5/\text{Si}-\text{Li}_{21}\text{Si}_5$ cell?
2. The heterogeneous lithiation of $\text{Si}-\text{Li}_{21}\text{Si}_5$ will lead to the SSE crack, but in $\text{Li}_{21}\text{Si}_5/\text{Si}-\text{Li}_{21}\text{Si}_5$, heterogeneous lithiation also exist in $\text{Si}-\text{Li}_{21}\text{Si}_5$. The only difference is the stress tolerance difference between SSE and $\text{Li}_{21}\text{Si}_5$. Therefore, the highest stress that SSE and $\text{Li}_{21}\text{Si}_5$ can withstand has to be provided.
3. The N/P ratio of all the full cells has to be provided.
4. In $\text{Li}_{21}\text{Si}_5/\text{Si}-\text{Li}_{21}\text{Si}_5$, the $\text{Li}_{21}\text{Si}_5$ layer contact with SSE will also provide Li^+ to Si in the $\text{Si}-\text{Li}_{21}\text{Si}_5$, what is structure evolution of $\text{Si}-\text{Li}_{21}\text{Si}_5$ anode with higher $\text{Li}_{21}\text{Si}_5$ content?
5. At the same lithiation time, the Si should have a lower or equal level of lithiation compared with that of $\text{Li}-\text{Si}@\text{Si}$ due to the lower Li^+ diffusion in Si. Without any constraint, the volume expansion of Si should be equal to or lower than that of $\text{Li}-\text{Si}@\text{Si}$. But in Figure 4a-b, the trend was opposite, more explanation has to be added.
6. Voltage profile has to be provided to explain the self-discharge.

Reviewer #3

(Remarks to the Author)

Zhang et al. showed a $\text{Li}_{21}\text{Si}_5/\text{Si}-\text{Li}_{21}\text{Si}_5$ double-layered anode design for silicon-based all-solid-state batteries (Si-ASSBs) operating free of external pressure. The authors claimed that $\text{Li}_{21}\text{Si}_5$ layer pressed upon $\text{Si}-\text{Li}_{21}\text{Si}_5$ served as a mixed ionic/electronic conductor layer, facilitating lithium transport to Si in the $\text{Si}-\text{Li}_{21}\text{Si}_5$ and inducing a uniform electric field at the anode interface. Accordingly, it was argued that the stress caused by volume expansion was released uniformly and the interface between the anode and the solid electrolyte was maintained, ensuring the bulk and interfacial stability of anode during cycling without operating pressure. However, although the authors proposed a potential anode design, supplementary experiments to strengthen the logic, appropriate interpretation of the results, and thoroughly polished writing

are highly required. We encourage that the authors address the following issues/concerns:

1. The authors emphasized the practicability of Li₂₁Si₅/Si-Li₂₁Si₅ anode design. However, it is not believed that the electrochemical test at 45 °C is consistent with the authors' claim. The electrochemical performance at room temperature is needed and needs to be compared with previous works of low-pressure operating ASSBs.
2. The lithium-ion flux indicated by the arrow in Figure 1 appears to represent the ion flux in the bulk solid electrolyte. It would be better to find a way to change the arrow position to the interface to express the ion flux at the interface.
3. What is the structure of Li₂₁Si₅-Si if the interface improvement with Li₂₁Si₅ layer is needed to homogenize the surface electric field? It is encouraged to check the results of Li₂₁Si₅-Si anode (morphological, electrochemical performances, etc.) and discuss the differences with Si-Li₂₁Si₅ and Li₂₁Si₅/Si-Li₂₁Si₅ anode designs.
4. The thickness of Si-Li₂₁Si₅ and Li₂₁Si₅ layers is 47.5 μm and 57.5 μm, which are considered too thick.
 - 1) It is recommended that the authors control the thickness, which is also related to N/P ratio control and investigate the effect of thickness (N/P ratio) on the cell performance.
 - 2) It would be a good study to calculate energy density with the current cell configuration and compare it to previous works.
5. In Figure S4, the meaning of the peaks at 92.5, 9.5 and 5.2 ppm should be further specified in terms of crystallinity and degree of silicon lithiation.
6. In Page 6 and 124th line, it is difficult to conclude that the electrodes sintered at high pressure are "efficient" ionic/electronic dual conductor or have "efficient" conductive networks only through the physicochemical and morphological characterizations. It is appropriate to emphasize "efficiency" in the description part of electrochemical performance.
7. The Li₂₁Si₅/Si-Li₂₁Si₅ showed the most uniform surface among the samples. Then, are there specific reasons why the roughness increases from 0.45 μm to 2.15 μm in Figure 2e-g even if more Li₂₁Si₅ was used in Si-Li₂₁Si₅ and Li₂₁Si₅/Si-Li₂₁Si₅ compared to pure Si.
8. In Figure 3a, the overall thickness appears to be maintained at ~50 μm after lithiation. How can the thickness be maintained even though the volume expands by 300 % during lithiation?
9. It is recommended to measure the diffusion coefficients of Li₂₁Si₅ as a reference and the diffusion coefficients of all the anodes at room temperature.
10. It is suggested to investigate the cell cycling performances at room temperature including the CCD test in symmetric cells, and rate capability and long-term cyclability in full cells.
11. In the discussion on page 11, 238th line, comparing the potential between the electrodes needs the voltage profiles of each anode.
12. In Figure 5 for ASSB cycling using mold cells, how did the authors achieve free external pressure? It is helpful to provide cell design information for better understanding and pressure monitoring data.
13. In Figure 5b, where does the large over potential at the beginning of 1st charging come from?
14. Is there a possible reason why the coulombic efficiency exceeds 100 % at 10.2 mA cm⁻² in Figure S11?
15. Correct figure revision and caption are needed for Figure 5f. It is confusing to understand what the inset means and the EIS information on the anodes after cycling.
16. To claim that the anode structure and its interface remain stable in Figure 5g, it is recommended to separate the resistances of the anode and cathode.
17. In Figure S12, the areal capacity is only 0.1mAh/cm² for cycling at 2C. This is barely meaningful for evaluating stability or activity.
18. Figure S14 requires an SEM image of the secondary electron mode for a clear cross section.
19. Items that need correction.
 - 1) Page 2, 61st line: lithium-ions → lithium ion
 - 2) Page 6, 140th line: a relatively stable a stable sulfide
 - 3) Page 7, 148th line: The areal capacity should be expressed as "mAh cm⁻²".
 - 4) Page 8, 168th line: charge-transfer resistance at SEI (RSEI) → ion-transfer resistance at SEI (RSEI)
 - 5) Page 9, 187th line: a "slow" increase in polarization → The authors seem to want to express "rapid" increase.
 - 6) Page 9, 195th line: micrometer → micrometer-sized
 - 7) Page 9, 197th line: TEM → The first notation of the abbreviation requires the full name of TEM.
 - 8) Page 11, 232nd line: HRTEM → The first notation of the abbreviation requires the full name of HRTEM.
 - 9) Figure 4h: The labeling of sample name needs to be changed from "Li₂₁Si₅/Si-Li₂₁Si₅ after cycling" to "Li₂₁Si₅ after cycling".

Reviewer #4

(Remarks to the Author)

Version 1:

Reviewer comments:

Reviewer #1

(Remarks to the Author)

The authors have conducted additional experiments to address my concerns. Unfortunately, it remains unclear what advantages this study offers. Most importantly, there is no direct evidence to support the authors' key claim that introducing an additional Li₂₁Si₅ interlayer enables operation without external pressure. I cannot support the publication of this paper

until the following issues are resolved:

1. The cell performances at different N/P ratios were shown in the revised paper. However, even at an N/P of 3.3, low reversible capacity and initial efficiency were observed, and the cycle performance at an N/P of 1.6 was poor. I think that the good performance at high N/P is due to the available Li source in the anode electrode. To stably operate at an areal capacity of 2.8 mAh/cm², it can be estimated that an anode thickness of over 80 μm seems to be required. Please present the achievable energy density through this cell design.
2. There is no data demonstrating that good cycle performance was achieved without external pressure. The authors should provide real-time pressure monitoring data at a high reversible capacity of 3.0 mAh/cm².
3. The expansion/contraction of Si during cycling is inevitable. Even if the additional Li₂₁Si₅ layer helps with uniform Li⁺ flux at the interface between the electrolyte layer and the anode electrode, the increase in interfacial resistance between silicon and the Li₂₁Si₅ matrix within the anode electrode could not be avoided. How to solve the expansion/contraction of Si during cycling?
4. In the authors' previous paper (EES, 2024, 17, 1061–1072), the areal capacity of 18 mAh/cm² with the high CE over 90 % was achieved even without a Li₂₁Si₅ protective layer. Then, with the addition of the Li₂₁Si₅ layer, is operation without external pressure possible at high capacities above 10 mAh/cm²?
5. When the volume ratio of Li₂₁Si₅ in the anode composite increases, can the same effect be expected as using an additional Li₂₁Si₅ interlayer.

Reviewer #2

(Remarks to the Author)

The revised manuscript can be accepted.

Reviewer #3

(Remarks to the Author)

The authors sought to address most of the issues related to the in-depth understanding of the Li₂₁Si₅/Si-Li₂₁Si₅ double-layered anode design. For reconsideration, we encourage the authors to address the following concerns further:

1. What pressure was used for the self-discharge test in Supplementary Figure 5? Sintering of Si/Li₂₁Si₅ in a lithium metal half cell under high pressure can cause lithium metal to creep throughout the SSE layer, leading to a short circuit. Please provide the test conditions in Figure or Experimental to avoid confusion.
2. It is suggested to switch the numbers between the two supplementary tables.

Version 2:

Reviewer comments:

Reviewer #1

(Remarks to the Author)

The authors have tried to address my concerns. Compared to the previous study, they claim that the use of the Li_{4.4}Si layer enables stable cycling without external pressure. However, the current Figure S14 does not adequately demonstrate stable cell operation under zero pressure. The authors should provide pressure monitoring data with charge-discharge profiles under stable cell operation conditions. If they can provide pressure monitoring data for 5 cycle at least, I would support publication of this paper.

Responses to Reviewers' comments

Reviewer #1:

This work introduces a concept for operating a stable all-solid-state battery without external pressure by forming a $\text{Li}_{21}\text{Si}_5/\text{Si-Li}_{21}\text{Si}_5$ bilayer. Although the proposed bilayer showed improved battery performance compared to the anode performance of a silicon-only or $\text{Li}_{21}\text{Si}_5$ single layer, the concept exhibits several technical limitations. As shown in the cross-sectional image in Figure 2, the thickness of the $\text{Li}_{21}\text{Si}_5$ used as a protective layer is 50 μm , which is not realistic. In fact, the thickness of the $\text{Si-Li}_{21}\text{Si}_5$ layer is also 50 μm , making the total anode layer thickness 100 μm . Considering the content of Si where the electrochemical reaction occurs, it is unclear what advantages can be gained, because the energy density in this cell configuration is expected to be very low. Additionally, there was no in-depth electrochemical property analysis about $\text{Li}_{21}\text{Si}_5$, such as its ionic conductivity and electrical conductivity. Furthermore, it is difficult to find any advancements in this paper compared to the authors' recently published work (Energy Environ. Sci., 2024, 17, 1061). To be considered for publication, a clear explanation of the new advancements and an in-depth additional analysis are required. As it stands, I cannot support the publication of this paper in Nature Communications.

Response: We highly appreciated the reviewer's work and professional comments. Thank you for your positive comments on our new concept for operating a stable all-solid-state battery without external pressure by forming a $\text{Li}_{21}\text{Si}_5/\text{Si-Li}_{21}\text{Si}_5$ bilayer, as well as the improved battery performance compared to the anode performance of a silicon-only or $\text{Li}_{21}\text{Si}_5$ single layer. The concern you raised is a technical limitation of this concept, that is the thickness of proposed $\text{Li}_{21}\text{Si}_5/\text{Si-Li}_{21}\text{Si}_5$ bilayer anode. The technical limitation can be successfully resolved simply by adjusting the mass of the composite powders:

The thickness of the bilayer composite anode can be reduced from 100.5 μm to 50.2 and 22.1 μm (Fig. R1). The N/P ratio can be controlled varies from 6.7 to 3.3 and 1.67, considering the cathode loading of 20 mg/cm^2 with an area capacity of 2.8

mAh/cm².

Figure R1. Cross-section SEM images of Li₂₁Si₅/Si-Li₂₁Si₅ anode with N/P ratios of 1.6 (a), 3.3 (b), 6.7 (c).

The cycling performance of Li₂₁Si₅/Si-Li₂₁Si₅ bilayer anode at different ratios are shown in Fig. R2. As the N/P ratio increased, the ICE of the cells exhibited values of 89.8%, 91.7%, and 96.3%, respectively, while the capacity retention after 100 cycles demonstrated values of 15.6%, 86.3%, and 88.2%, respectively. These findings indicate that saturated lithiation of Si directly results in low ICE and capacity decay of the cells during cycling operating free from external pressure.

Figure R2. Cycle performance of Li₂₁Si₅/Si-Li₂₁Si₅-ASSBs with different N/P ratio.

Figure R3. Cross-section SEM images of Li₂₁Si₅/Si-Li₂₁Si₅ anode with N/P ratios of 1.6 (a), 3.3 (b), 6.7 (c) after cycling.

After cycling, the thickness evolution of Li₂₁Si₅/Si-Li₂₁Si₅ bilayer anodes with different thickness were also evaluated. Fig. R3 displays the cross-section SEM images of above Li₂₁Si₅/Si-Li₂₁Si₅ anode after cycling with thicknesses of 37.4, 64.9, and 121.4 μm, corresponding to expansion rate of 69.2%, 29.3%, and 15.6%,

respectively. It is evident that the electrode with N/P ratios of 1.6 exhibit pronounced cracking, which is likely the underlying cause of their performance degradation. Indeed, the majority of commercially available Si anode are designed to circumvent the phenomenon of saturated lithiation of Si.

Though the energy density of the solid-state cell could reach a breakthrough at the current stage, the design concept indeed makes a big progress on the application of solid-state batteries operated free from external pressure.

The main novelty and scientific contribution of this work are the unique bilayer anode design and stable cycling without external pressure, which was enabled by the uniform and homogeneous electric field induced by $\text{Li}_{21}\text{Si}_5$ interlayer. In our previous published work (Energy Environ. Sci., 2024, 17, 1061), we have successfully prepared Li-Si alloys and optimized the ratio of Li-Si, demonstrating the excellent electrical conductivity of $\text{Li}_{21}\text{Si}_5$ alloys, which enable ASSBs cycling under high pressure. However, the Si- $\text{Li}_{21}\text{Si}_5$ anode was unable to cycling operating free from external pressure, even at an N/P ratio of 6.3, due to the risk of a soft short circuit. This is a serious challenge for its further development. In light of these findings, this manuscript focuses on the heterogeneous lithiation and soft short-circuiting of Si- $\text{Li}_{21}\text{Si}_5$, conducting in-depth studies on its expansion stress release mechanism, interface, and ion/electron transport mechanism in the bulk phase. This has led to the development of a novel bilayer anode ($\text{Li}_{21}\text{Si}_5/\text{Si}$ - $\text{Li}_{21}\text{Si}_5$ anode), which has enabled the Si-ASSBs operating free from external pressure.

In order to in-depth further analyze the distribution of the current and stress, we added simulation experiments at the interface (the lower the N/P ratio, the more the Si expansion stress when the cathode capacity is kept constant).

Stress simulation under different electric fields

Figure R4. Current and stress distribution of Si/SSE (a) and Si/Li₂₁Si₅/SSE (b) models during heterogeneous lithiation.

As shown in Fig. R4a and b, we simplified the Si-Li₂₁Si₅ anode as a Si anode, established two models (Si/SSE and Si/Li₂₁Si₅/SSE), and calculated the current and stress distributions during the lithiation process based on finite element simulations. The results demonstrate that during heterogeneous lithiation of the Si/SSE model, the Li_xSi ($x < 3.75$) generated at the interface between the Si and SSE exhibits enhanced electrical conductivity, which attracts the lithium ions and causes a marked increase in expansion stress at the interfaces. In contrast, when the Si/Li₂₁Si₅/SSE model undergoes heterogeneous lithiation, the superior conductivity of Li₂₁Si₅ ensures that the lithium ions are always in a homogenized electric field, resulting in a significantly smaller stress at the interface compared to the Si/SSE model.

Reviewer #2:

The stable operation of the Si anode at low pressure is important for the commercialization of all-solid-state cells. The following comments have to be addressed before paper acceptance.

Response: Thank you for the positive recommendation on our paper. We acknowledge reviewer for their constructive comments. The manuscript has been revised accordingly with point-by-point responses to the comments.

1. What is the impact of Si/Li₂₁Si₅ weight ratio in Si-Li₂₁Si₅ on the electrochemical performance of Li₂₁Si₅/Si-Li₂₁Si₅ cell?

Response: We are grateful and acknowledge the reviewer for the valuable comment. Based on our previous work (Energy Environ. Sci., 2024, 17, 1061), the optimization of the ratio of the anode can have a marked effect on the bulk-phase conductivity of the Si-Li₂₁Si₅ anode, which in turn exerts a significant influence on the ICE and the cycling stability of the cells.

(Figure 5 cited in Energy Environ. Sci., 2024, 17, 1061).

Figure 5. The effect of Li₂₁Si₅ alloy content on electrochemical characteristics of ASSBs. a-c, first charge-discharge cycle, ICE, and 1C short-cycle curves of Si/Li₂₁Si₅-ASSBs with different Li₂₁Si₅ mass percentages (0 wt%, 25 wt%, 50 wt%, 75 wt%, 100 wt%). d, the DC polarization of Si/Li₂₁Si₅ electrodes with 0 wt%, 50 wt% and 100 wt% of Li₂₁Si₅. e, lithium diffusion coefficient of Si/Li₂₁Si₅-ASSBs with 0 wt%, 50 wt% and 100 wt% of Li₂₁Si₅ powder calculated from PITT.

As shown in Figure 5, to determine the optimal content of Li₂₁Si₅ alloy in the Si/Li₂₁Si₅ anodes, pure Si (0 wt%), pure Li₂₁Si₅ (100 wt%) and Si/Li₂₁Si₅ anodes with different Li₂₁Si₅ mass percentages (25 wt%, 50 wt%, 75 wt%) were used to assemble the ASSBs and cycled under high pressure. The first charge–discharge curves and ICEs are shown in Fig. 5a and b. During the charging process, compared with the pure Si anode, the Si/Li₂₁Si₅ anodes caused a decrease of 0.1–0.2 V of the charging plateau. The polarization was also significantly reduced. When the content of Li₂₁Si₅ reached 100 wt% in the electrode, the charging plateau rapidly rose to 3.95 V, similar to that of lithium-anoded ASSBs, indicating the occurrence of Li metal deposition. At

the end of discharge, the ASSBs with different $\text{Li}_{21}\text{Si}_5$ contents exhibited discharge specific capacities of 72.1 mA h g^{-1} , $137.2 \text{ mA h g}^{-1}$, $140.6 \text{ mA h g}^{-1}$, $136.8 \text{ mA h g}^{-1}$, and $139.1 \text{ mA h g}^{-1}$, with ICEs of 51.4%, 96.7%, 97.8%, 97.4%, and 97.3%. This indicates that the addition of $\text{Li}_{21}\text{Si}_5$ not only improved the conductivity of the electrode, but also provides a rich lithium source for super high ICEs, which will benefit the subsequent cycling performance of the battery.

Fig. 5c illustrates the 1C short-cycle curves for the above five cells. At a current density of 3 mA cm^{-2} , the performance of ASSBs with a pure Si anode showed a rapid decline and failure. The performance of the ASSB with 100 wt% $\text{Li}_{21}\text{Si}_5$ showed slower degradation, with only 58.6 mA h g^{-1} of the specific capacity retained after 10 cycles (62.3% capacity retention), which was probably due to the growth of lithium dendrites. As a comparison, the Si/ $\text{Li}_{21}\text{Si}_5$ electrodes with $\text{Li}_{21}\text{Si}_5$ contents of 25–75 wt% showed ultra-high cycling stability, with specific capacities between 80.0 mA h g^{-1} and $108.2 \text{ mA h g}^{-1}$. Among them, the Si/ $\text{Li}_{21}\text{Si}_5$ -ASSBs with 50 wt% $\text{Li}_{21}\text{Si}_5$ exhibited the best reversible specific capacity.

The DC polarization of the electrodes with 0 wt%, 50 wt%, and 100 wt% $\text{Li}_{21}\text{Si}_5$ is illustrated in Fig. 5d. The current responses of the electrodes with 50 wt% and 100 wt% $\text{Li}_{21}\text{Si}_5$ were almost the same and much higher than that of the pure Si anode. And their electronic conductivities were 0.002 S cm^{-1} , 0.26 S cm^{-1} , and 0.34 S cm^{-1} , respectively, indicating excellent electron transport paths constructed by the addition of 50 wt% $\text{Li}_{21}\text{Si}_5$. Furthermore, Fig. 5e show the lithium diffusion coefficient of the above Si-ASSBs calculated from PITT. The Li^+ coefficient of LCO | SE | Si/ $\text{Li}_{21}\text{Si}_5$ with 50 wt% $\text{Li}_{21}\text{Si}_5$ was $1.47 \times 10^{-6} \text{ cm}^2 \text{ S}^{-1}$, and was far superior to $0.58 \times 10^{-6} \text{ cm}^2 \text{ S}^{-1}$ for LCO | SE | Si. The superior Li^+ coefficient benefits the processes of ion diffusion from the cathode to the anode and alloying with Si.

Therefore, in this work, the Si/ $\text{Li}_{21}\text{Si}_5$ electrode with 50 wt% $\text{Li}_{21}\text{Si}_5$ was used as the anode for Si/ $\text{Li}_{21}\text{Si}_5$ -ASSBs operating free from external pressure. The results demonstrate that the ASSB's performance is significantly diminished when operating under high-pressure. At other ratios, the aforementioned issues would be exacerbated.

Revised parts in the manuscript:

In page 4, “Based on our previous work, [22] the $\text{Li}_{21}\text{Si}_5$ powder was first prepared by a spontaneous Li–Si alloying method, followed by its mixing with Si particles by stack pressure (50 wt% of $\text{Li}_{21}\text{Si}_5$ alloy was determined to be the optimal ratio for the anode).”

2. The heterogeneous lithiation of Si- $\text{Li}_{21}\text{Si}_5$ will lead to the SSE crack, but in $\text{Li}_{21}\text{Si}_5/\text{Si}$ - $\text{Li}_{21}\text{Si}_5$, heterogeneous lithiation also exist in Si- $\text{Li}_{21}\text{Si}_5$. The only difference is the stress tolerance difference between SSE and $\text{Li}_{21}\text{Si}_5$. Therefore, the highest stress that SSE and $\text{Li}_{21}\text{Si}_5$ can withstand has to be provided.

Response: We are grateful and acknowledge the reviewer for the professional comments. In order to further analyze the distribution of stress and the maximum value that can be withstood, we added simulation experiments and performance tests at different N/P ratios (the lower the N/P ratio, the more the Si expansion stress when the cathode capacity is kept constant).

Stress simulation under different electric fields

Figure R1. Current and stress distribution of Si/SSE (i) and Si/ $\text{Li}_{21}\text{Si}_5$ /SSE (j) models during heterogeneous lithiation.

As shown in Figure R1a and b, we simplified the Si- $\text{Li}_{21}\text{Si}_5$ anode as a Si anode, established two models (Si/SSE and Si/ $\text{Li}_{21}\text{Si}_5$ /SSE), and calculated the current and stress distributions during the lithiation process based on finite element simulations. The results demonstrate that during heterogeneous lithiation of the Si/SSE model, the Li_xSi ($x < 3.75$) generated at the interface between the Si and SSE exhibits enhanced electrical conductivity, which attracts the lithium ions and causes a marked increase in expansion stress at the interfaces. In contrast, when the Si/ $\text{Li}_{21}\text{Si}_5$ /SSE model undergoes heterogeneous lithiation, the superior conductivity of $\text{Li}_{21}\text{Si}_5$ ensures that

the lithium ions are always in a homogenized electric field, resulting in a significantly smaller stress at the interface compared to the Si/SSE model.

Furthermore, we prepared electrodes with different thicknesses and tested the cell cycling performance with N/P ratios of 6.7, 3.3 and 1.6. With the same cathode mass loading, the anode with a low N/P ratio lithiated more completely and the expansion stress at the interface was greater.

The thickness of the bilayer composite anode can be reduced from 100.5 μm to 50.2 and 22.1 μm (Fig. R2). The N/P ratio can be controlled varies from 6.7 to 3.3 and 1.67, considering the cathode loading of 20 mg/cm^2 with an area capacity of 2.8 mAh/cm^2 .

Figure R2. Cross-section SEM images of $\text{Li}_{21}\text{Si}_5/\text{Si-Li}_{21}\text{Si}_5$ anode with N/P ratios of 1.6 (a), 3.3 (b), 6.7 (c).

Figure R3. Cycle performance of $\text{Li}_{21}\text{Si}_5/\text{Si-Li}_{21}\text{Si}_5$ -ASSBs with different N/P ratio.

Figure R4. Cross-section SEM images of $\text{Li}_{21}\text{Si}_5/\text{Si-Li}_{21}\text{Si}_5$ anode with N/P ratios of 1.6 (a), 3.3 (b), 6.7 (c) after cycling.

The cycling performance of $\text{Li}_{21}\text{Si}_5/\text{Si-Li}_{21}\text{Si}_5$ bilayer anode at different ratios are shown in Fig. R3. As the N/P ratio increased, the ICE of the cells exhibited values of 89.8%, 91.7%, and 96.3%, respectively, while the capacity retention after 100 cycles demonstrated values of 15.6%, 86.3%, and 88.2%, respectively. These findings indicate that saturated lithiation of Si directly results in low ICE and capacity decay of the cells during cycling operating free from external pressure

After cycling, the thickness evolution of $\text{Li}_{21}\text{Si}_5/\text{Si-Li}_{21}\text{Si}_5$ bilayer anodes with different thickness were also evaluated. Fig. R4 displays the cross-section SEM images of above $\text{Li}_{21}\text{Si}_5/\text{Si-Li}_{21}\text{Si}_5$ anode after cycling with thicknesses of 37.4, 64.9, and 121.4 μm , corresponding to expansion rate of 69.2%, 29.3%, and 15.6%, respectively. It is evident that the electrode with N/P ratios of 1.6 exhibit pronounced cracking, which is likely the underlying cause of their performance degradation.

In conclusion, when the N/P ratio is 1.6, the expansion stress generated by the anode will exceed the stress tolerance of the interface, leading to performance degradation.

Revised parts in the manuscript:

In page 4, “5 mg $\text{Si-Li}_{21}\text{Si}_5$ powders with 50 wt% (a_1), 75 wt% (a_2) and 100 wt% (a_3) of $\text{Li}_{21}\text{Si}_5$ was respectively weighed into the mould of ASSBs and cold-pressed under 600 MPa for 3 min to obtain electrode. As shown in Fig. 2a₁-a₃, their thicknesses were 22.9, 22.1 and 21.6 μm , respectively. As the $\text{Li}_{21}\text{Si}_5$ alloy content increased, the electrodes became denser higher. And then, $\text{Li}_{21}\text{Si}_5$ electrode was pressed on the $\text{Si-Li}_{21}\text{Si}_5$ electrode (50 wt% of $\text{Li}_{21}\text{Si}_5$) under a stack pressure of 600 MPa, thus forming a $\text{Li}_{21}\text{Si}_5/\text{Si-Li}_{21}\text{Si}_5$ anode with the thickness of 50.2 μm (Fig. 2a₄). Further increasing the powder mass to 10 mg enabled the fabrication of electrodes with a thickness of 105 μm , in which the $\text{Si-Li}_{21}\text{Si}_5$ and $\text{Li}_{21}\text{Si}_5$ layers are 47.5 μm and 57.5 μm thick, respectively. In Fig. 2a₂, a₄, and a₅, the theoretical maximum areal capacity of the $\text{Li}_{21}\text{Si}_5/\text{Si-Li}_{21}\text{Si}_5$ anode is 4.67, 9.37, and 18.75 mAh cm^{-2} , respectively.”

In page 7, “In order to further analyze the distribution of current and stress, we simplified the $\text{Si-Li}_{21}\text{Si}_5$ anode as a Si anode, established two models (Si/SSE and

Si/Li₂₁Si₅/SSE) via COMSOL, and calculated the current and stress distributions during the lithiation process based on finite element simulations. As shown in Fig. 2i and j, the results demonstrate that during heterogeneous lithiation of the Si/SSE model, the Li_xSi (x<3.75) generated at the interface between the Si and SSE exhibits enhanced electrical conductivity, which attracts the lithium ions and causes a marked increase in expansion stress at the interfaces. In contrast, when the Si/Li₂₁Si₅/SSE model undergoes heterogeneous lithiation, the superior conductivity of Li₂₁Si₅ ensures that the lithium ions are always in a homogenized electric field, resulting in a significantly smaller stress at the interface compared to the Si/SSE model.”

In page 20-21, “The finite element analysis based on the commercial software COMSOL Multiphysics was adopted by the method previously reported in the research. ^[42] In Fig.2 i and j, a periodicity simplified 2D structure was established to simulate the phase transition due to the chemical/electrochemical reaction caused by lithium ions and electron transport in materials.

The current density (derived from Li-ion transport) of this model is determined by Ohm's law:

$$j_s = -\sigma_s \nabla \varphi_s$$

Where j_s is current density and φ_s is potential distribution of this model, σ_s is the resistance which is related to electronic and ionic conductivity of material.

To simulate the phenomenon of phase interface transition associated with the process of charge transfer (reaction between lithium ions and Si), we used the simplified Navier-Stokes equation, called Level-Set, to describe the whole system as following expression shown:

$$\frac{\partial \phi}{\partial t} + u \cdot \nabla \phi = \gamma \nabla \cdot \left[\varepsilon \nabla \phi - \phi(1 - \phi) \frac{\nabla \phi}{|\nabla \phi|} \right]$$

Where γ is a reinitialization parameter to assist equation solved easily, ε is parameter controlling interface thickness to describe the thickness of phase interface, u is the velocity field of phase interface. ϕ is used to describe the phase state of material, and it is defined as:

$$\phi = \begin{cases} 0, & \text{Reacted Si} \\ 1, & \text{Si} \end{cases}$$

One should note that phase transition is controlled by velocity field u , which is related to the current j_s in Si, and we have,

$$u = \frac{j_s M_{Li}}{F \rho}$$

Where M_{Li} is the relative atomic mass of lithium, F is the Faraday constant, ρ is the mass participating in the Si reaction per unit volume. We then use the previous study to solve the stress-strain simulation of the modified system. [42]”

Reference

42. Luo, L. et al. Insights into the enhanced interfacial stability enabled by electronic conductor layers in solid-state Li batteries. *Adv. Energy Mater.* **13**, 2203517 (2023).

In page 15, “To analyze the effect of the lithiation depth of Si in the $\text{Li}_{21}\text{Si}_5/\text{Si-Li}_{21}\text{Si}_5$ anode on its cycling performance, as shown in Fig. 5e, ASSBs with N/P ratios of 1.6, 3.3, and 6.7 were tested at the same current density and cathode capacity mass loading. As the N/P ratio increased, the ICE of the cells exhibited values of 89.8%, 91.7%, and 96.3%, respectively, while the capacity retention after 100 cycles demonstrated values of 15.6%, 86.3%, and 88.2%, respectively. These findings indicate that saturated lithiation of Si directly results in low ICE and capacity decay of the cells during cycling operating free from external pressure. Fig. S14 displays the cross-section SEM images of above $\text{Li}_{21}\text{Si}_5/\text{Si-Li}_{21}\text{Si}_5$ anode after cycling with thicknesses of 37.4, 64.9, and 121.4 μm , corresponding to expansion rate of 69.2%, 29.3%, and 15.6%, respectively. It is evident that the electrode with N/P ratios of 1.6 exhibit pronounced cracking, which is likely the underlying cause of their performance degradation. Indeed, the majority of commercially available Si anode are designed to circumvent the phenomenon of saturated lithiation of Si.”

3. The N/P ratio of all the full cells has to be provided.

Response: We express thanks to reviewer for the valuable comment. We have added the relevant N/P ratio descriptions in the revised manuscript. In addition, we have

added performance tests and structural characterization for different N/P ratios (refer to the second review comment).

Revised parts in the manuscript:

In page 8, “It should be noted that the $\text{Li}_{21}\text{Si}_5/\text{Si-Li}_{21}\text{Si}_5$ anode has a clearer bilayer structure at a high N/P ratio. In order to show the comparative performance of different electrodes, the electrochemical performance tests of the $\text{Li}_{21}\text{Si}_5/\text{Si-Li}_{21}\text{Si}_5$ anode in the following section were carried out based on an N/P ratio of 6.7.”

In page 13, “At this time, their N/P ratios were 11, 3.5, and 2.4, with discharge capacities of 1.7 mAh cm^{-2} , 5.4 mAh cm^{-2} , and 7.8 mAh cm^{-2} , respectively.”

In page 14, “(d), Rate capability test. (e) Cycle performance of $\text{Li}_{21}\text{Si}_5/\text{Si-Li}_{21}\text{Si}_5$ -ASSBs with different N/P ratios. (f) Long cycle performance of $\text{Li}_{21}\text{Si}_5/\text{Si-Li}_{21}\text{Si}_5$ -ASSBs (N/P=6.7).”

4. In $\text{Li}_{21}\text{Si}_5/\text{Si-Li}_{21}\text{Si}_5$, the $\text{Li}_{21}\text{Si}_5$ layer contact with SSE will also provide Li^+ to Si in the $\text{Si-Li}_{21}\text{Si}_5$, what is structure evolution of $\text{Si-Li}_{21}\text{Si}_5$ anode with higher $\text{Li}_{21}\text{Si}_5$ content?

Response: Thanks for the constructive suggestion by the referees. When the $\text{Li}_{21}\text{Si}_5$ alloy content is higher, as shown in Fig. 2a2 (in the revised manuscript), the cold-press sintering of the anode will be more thorough as Si is completely surrounded by $\text{Li}_{21}\text{Si}_5$. At this point, the voltage at the anode is further close to 0 V, making it more susceptible to soft short circuits. This situation is analyzed in our published work ^[1] (refer to the first review comment).

Reference

[1] Zhang, Z. et al. An all-electrochem-active silicon anode enabled by spontaneous Li–Si alloying for ultra-high performance solid-state batteries. *Energy Environ. Sci.* 17, 1061–1072 (2024).

5. At the same lithiation time, the Si should have a lower or equal level of lithiation

compared with that of Li-Si@Si due to the lower Li⁺ diffusion in Si. Without any constraint, the volume expansion of Si should be equal to or lower than that of Li-Si@Si. But in Figure 4a-b, the trend was opposite, more explanation has to be added.

Response: We acknowledge good suggestion by the reviewer. Since the lithium ions diffusion of Si is lower than that of Li-Si@Si, the Si anode requires a longer time to activate. As shown in Fig. 4a-b, the Li-Si@Si anode started to expand after 60 s and the lithiation was completed after 70 s. In contrast, the Si anode started to expand after 160 s and the lithiation was completed after 200 s. In the actual low-rate lithiation process, both the Si anode and the Li-Si@Si anode have enough time to react. Therefore, in In-situ TEM test, we allowed enough time for lithiation. Benefiting from the pre-lithiation of the Li₂₁Si₅/Si-Li₂₁Si₅ anode during the cold-pressed sintering process, part of Li-Si@Si's volume expansion stress had been released in advance, while the subsequent uniform lithiation avoided fragmentation of the Si particles. As a result, the Li-Si@Si anode has a lower expansion rate. To enhance the quality of the article, we have added appropriate explanations in the revised manuscript.

Revised parts in the manuscript:

In page 11, “In the actual low-rate lithiation process, both the Si anode and the Li-Si@Si anode have enough time to react. Therefore, the time of lithiation is not limited in the in-situ TEM test.”

In page 12, “Benefiting from the pre-lithiation of the Li₂₁Si₅/Si-Li₂₁Si₅ anode during the cold-pressed sintering process, part of Li-Si@Si's volume expansion stress had been released in advance, while the subsequent uniform lithiation avoided fragmentation of the Si particles. As a result, the Li-Si@Si anode has a lower expansion rate.”

6. Voltage profile has to be provided to explain the self-discharge.

Response: Thank you very much for your thoughtful and valuable comments. We

have added voltage profiles during self-discharge to the revised manuscript. As shown in Fig. R, after Si and $\text{Li}_{21}\text{Si}_5$ is co-pressed of $\text{Si-Li}_{21}\text{Si}_5$, lithium ions in the $\text{Li}_{21}\text{Si}_5$ alloy continue to enter the bulk phase of Si to form Li_xSi , as indicated by the voltage at the anode decreases continuously as the value of “x” increases.

Figure R. Voltage profile during a long cold-pressed sintering process.

Revised parts in the manuscript:

In page 6, “As shown in Fig. S5, when $\text{Si-Li}_{21}\text{Si}_5$ is cold-sintered for a long time, lithium ions in the $\text{Li}_{21}\text{Si}_5$ alloy continue to enter the bulk phase of Si to form Li_xSi . The voltage at the anode decreases continuously as the value of “x” increases.”

Reviewer #3:

Zhang et al. showed a $\text{Li}_{21}\text{Si}_5/\text{Si-Li}_{21}\text{Si}_5$ double-layered anode design for silicon-based all-solid-state batteries (Si-ASSBs) operating free of external pressure. The authors claimed that $\text{Li}_{21}\text{Si}_5$ layer pressed upon Si- $\text{Li}_{21}\text{Si}_5$ served as a mixed ionic/electronic conductor layer, facilitating lithium transport to Si in the Si- $\text{Li}_{21}\text{Si}_5$ and inducing a uniform electric field at the anode interface. Accordingly, it was argued that the stress caused by volume expansion was released uniformly and the interface between the anode and the solid electrolyte was maintained, ensuring the bulk and interfacial stability of anode during cycling without operating pressure. However, although the authors proposed a potential anode design, supplementary experiments to strengthen the logic, appropriate interpretation of the results, and thoroughly polished writing are highly required. We encourage that the authors address the following issues/concerns:

Response: We are grateful and acknowledge the reviewer for the positive comments. The manuscript has been revised accordingly with point-by-point responses to the comments.

1. The authors emphasized the practicability of $\text{Li}_{21}\text{Si}_5/\text{Si-Li}_{21}\text{Si}_5$ anode design. However, it is not believed that the electrochemical test at 45 °C is consistent with the authors' claim. The electrochemical performance at room temperature is needed and needs to be compared with previous works of low-pressure operating ASSBs.

Response: This is a constructive suggestion, and we acknowledge this comment from the reviewer. As shown in Fig. Ra and b, the $\text{Li}_{21}\text{Si}_5/\text{Si-Li}_{21}\text{Si}_5$ -ASSB can be cycled at room temperature, but the synthetic electrolyte has a low ionic conductivity of $8.46 \times 10^{-4} \text{ S cm}^{-1}$ at room temperature, with the higher ionic conductivity of $1.87 \times 10^{-3} \text{ S cm}^{-1}$ at 45 °C (the area of the electrolyte sheet is 0.785 cm^2). However, its low ionic conductivity at 25 °C resulted in a low average CE (99.54%) of ASSB that leads to poor performance, and the capacity retention of only 80% after 45 cycles. Nonetheless, the core of our article is to highlight the $\text{Li}_{21}\text{Si}_5/\text{Si-Li}_{21}\text{Si}_5$ anode design in ASSBs. By moderately increasing the temperature, it is possible to enhance the lithium ions transport efficiency of ASSB, thereby demonstrating the advanced nature of $\text{Li}_{21}\text{Si}_5/\text{Si-Li}_{21}\text{Si}_5$ anode design.

Figure R. (a) Nyquist plots of 80 mg bilayer electrolyte at different temperatures. (b) Cycle performance of $\text{Li}_{21}\text{Si}_5/\text{Si-Li}_{21}\text{Si}_5$ -ASSBs at 25 °C.

2. The lithium-ion flux indicated by the arrow in Figure 1 appears to represent the ion flux in the bulk solid electrolyte. It would be better to find a way to change the arrow position to the interface to express the ion flux at the interface.

Response: We acknowledge good suggestion by the reviewer. We extend our sincerest apologies for the confusion. To clarify, we have included an illustration of the arrows in Figure 1.

3. What is the structure of $\text{Li}_{21}\text{Si}_5$ -Si if the interface improvement with $\text{Li}_{21}\text{Si}_5$ layer is needed to homogenize the surface electric field? It is encouraged to check the results of $\text{Li}_{21}\text{Si}_5$ -Si anode (morphological, electrochemical performances, etc.) and discuss the differences with $\text{Si-Li}_{21}\text{Si}_5$ and $\text{Li}_{21}\text{Si}_5/\text{Si-Li}_{21}\text{Si}_5$ anode designs.

Response: Thank you very much for your thoughtful and valuable comments. The structural morphology and electrochemical properties of $\text{Si-Li}_{21}\text{Si}_5$ are discussed in detail in our previously published article and thus not repeated in this manuscript (Energy Environ. Sci., 2024, 17, 1061). To better demonstrate the structure, we have included relevant descriptions and references in the revised manuscript.

This manuscript focuses on the differences between the $\text{Si-Li}_{21}\text{Si}_5$ and $\text{Li}_{21}\text{Si}_5/\text{Si-Li}_{21}\text{Si}_5$ anode designs operating free from external pressure, and the variations of $\text{Si-Li}_{21}\text{Si}_5$ have been previously discussed in Fig. 1b, Fig. 2a, Fig. 2d, Fig. 2f, Fig. 3b, Fig. 3g, Fig. S15b, Fig. 6h-i, and Fig. S17 of the manuscript. To further

elucidate the distinctions between the two, we have conducted a morphological characterization and electrochemical performance assessment of the $\text{Li}_{21}\text{Si}_5/\text{Si-Li}_{21}\text{Si}_5$ anode with different N/P ratios. The comprehensive findings can be found in the fourth review comment.

Revised parts in the manuscript:

In page 4, “Based on our previous work,^[22] the $\text{Li}_{21}\text{Si}_5$ powder was first prepared by a spontaneous Li–Si alloying method, followed by its mixing with Si particles by stack pressure (50 wt% of $\text{Li}_{21}\text{Si}_5$ alloy was determined to be the optimal ratio for the anode).”

In page 5-6, “The surface SEM image of $\text{Si-Li}_{21}\text{Si}_5$ electrode with 50 wt% $\text{Li}_{21}\text{Si}_5$ is shown in the Fig. 2d, the $\text{Li}_{21}\text{Si}_5$ particles (black) were distributed in a uniform manner around the Si particles (white), forming an excellent three-dimensional continuous conductive network enabled by cold-pressed sintering.”

4. The thickness of $\text{Si-Li}_{21}\text{Si}_5$ and $\text{Li}_{21}\text{Si}_5$ layers is 47.5 μm and 57.5 μm , which are considered too thick.

1) It is recommended that the authors control the thickness, which is also related to N/P ratio control and investigate the effect of thickness (N/P ratio) on the cell performance.

Response: We acknowledge reviewer for the valuable comments. We prepared electrodes with different thicknesses and tested the cell cycling performance with N/P ratios.

The thickness of the bilayer composite anode can be reduced from 100.5 μm to 50.2 and 22.1 μm (Fig. R1). The N/P ratio can be controlled varies from 6.7 to 3.3 and 1.67, considering the cathode loading of 20 mg/cm^2 with an area capacity of 2.8 mAh/cm^2 .

Figure R1. Cross-section SEM images of $\text{Li}_{21}\text{Si}_5/\text{Si-Li}_{21}\text{Si}_5$ anode with N/P ratios of 1.6 (a), 3.3 (b), 6.7 (c).

The cycling performance of $\text{Li}_{21}\text{Si}_5/\text{Si-Li}_{21}\text{Si}_5$ bilayer anode at different ratios are shown in Fig. R2. As the N/P ratio increased, the ICE of the cells exhibited values of 89.8%, 91.7%, and 96.3%, respectively, while the capacity retention after 100 cycles demonstrated values of 15.6%, 86.3%, and 88.2%, respectively. These findings indicate that saturated lithiation of Si directly results in low ICE and capacity decay of the cells during cycling operating free from external pressure.

Figure R2. Cycle performance of $\text{Li}_{21}\text{Si}_5/\text{Si-Li}_{21}\text{Si}_5$ -ASSBs with different N/P ratio.

Figure R3. Cross-section SEM images of $\text{Li}_{21}\text{Si}_5/\text{Si-Li}_{21}\text{Si}_5$ anode with N/P ratios of 1.6 (a), 3.3 (b), 6.7 (c) after cycling.

After cycling, the thickness evolution of $\text{Li}_{21}\text{Si}_5/\text{Si-Li}_{21}\text{Si}_5$ bilayer anodes with different thickness were also evaluated. Fig. R3 displays the cross-section SEM images of above $\text{Li}_{21}\text{Si}_5/\text{Si-Li}_{21}\text{Si}_5$ anode after cycling with thicknesses of 37.4, 64.9, and 121.4 μm, corresponding to expansion rate of 69.2%, 29.3%, and 15.6%, respectively. It is evident that the electrode with N/P ratios of 1.6 exhibit pronounced

cracking, which is likely the underlying cause of their performance degradation. Indeed, the majority of commercially available Si anode are designed to circumvent the phenomenon of saturated lithiation of Si.

Revised parts in the manuscript:

In page 4, “5 mg Si-Li₂₁Si₅ powders with 50 wt% (a₁), 75 wt% (a₂) and 100 wt% (a₃) of Li₂₁Si₅ was respectively weighed into the mould of ASSBs and cold-pressed under 600 MPa for 3 min to obtain electrode. As shown in Fig. 2a₁-a₃, their thicknesses were 22.9, 22.1 and 21.6 μm, respectively. As the Li₂₁Si₅ alloy content increased, the electrodes became denser higher. And then, Li₂₁Si₅ electrode was pressed on the Si-Li₂₁Si₅ electrode (50 wt% of Li₂₁Si₅) under a stack pressure of 600 MPa, thus forming a Li₂₁Si₅/Si-Li₂₁Si₅ anode with the thickness of 50.2 μm (Fig. 2a₄). Further increasing the powder mass to 10 mg enabled the fabrication of electrodes with a thickness of 105 μm, in which the Si-Li₂₁Si₅ and Li₂₁Si₅ layers are 47.5 μm and 57.5 μm thick, respectively. In Fig. 2a₂, a₄, and a₅, the theoretical maximum areal capacity of the Li₂₁Si₅/Si-Li₂₁Si₅ anode is 4.67, 9.37, and 18.75 mAh cm⁻², respectively.”

In page 15, “To analyze the effect of the lithiation depth of Si in the Li₂₁Si₅/Si-Li₂₁Si₅ anode on its cycling performance, as shown in Fig. 5e, ASSBs with N/P ratios of 1.6, 3.3, and 6.7 were tested at the same current density and cathode capacity mass loading. As the N/P ratio increased, the ICE of the cells exhibited values of 89.8%, 91.7%, and 96.3%, respectively, while the capacity retention after 100 cycles demonstrated values of 15.6%, 86.3%, and 88.2%, respectively. These findings indicate that saturated lithiation of Si directly results in low ICE and capacity decay of the cells during cycling operating free from external pressure. Fig. S14 displays the cross-section SEM images of above Li₂₁Si₅/Si-Li₂₁Si₅ anode after cycling with thicknesses of 37.4, 64.9, and 121.4 μm, corresponding to expansion rate of 69.2%, 29.3%, and 15.6%, respectively. It is evident that the electrode with N/P ratios of 1.6 exhibit pronounced cracking, which is likely the underlying cause of their performance degradation. Indeed, the majority of commercially available Si anode are designed to circumvent

the phenomenon of saturated lithiation of Si.”

2) It would be a good study to calculate energy density with the current cell configuration and compare it to previous works.

Response: We acknowledge reviewer for the valuable comments. The cycling of ASSBs operating free from external pressure presents a significant challenge, and in this manuscript, we focus on the design of the anode. In order to minimize the impact of the SSE and cathode, a double-layer SSE and a lower energy density cathode (LCO) were employed due to their relatively more stable performance. It is inevitable that such a strategy will result in a ASSB with a lower energy density.

Nevertheless, it is our contention that the integration of thin SSE and high-specific-capacity cathodes (e.g. NCM) through dry electrode process and coating strategies can serve to enhance the energy density of ASSBs. We would like to express our gratitude to the reviewers for their constructive comments, which will encourage our future research and development in this area.

5. In Figure S4, the meaning of the peaks at 92.5, 9.5 and 5.2 ppm should be further specified in terms of crystallinity and degree of silicon lithiation.

Response: We acknowledge good suggestion by the reviewer. The 92.5 ppm peak indicates the fully lithiated silicon in the form of $\text{Li}_{21}\text{Si}_5$. The ^7Li NMR peaks around 9.5 ppm and 5.2 ppm can be assigned to phases of $\text{Li}_{15}\text{Si}_4$ and $\text{Li}_{13}\text{Si}_4$, respectively. The meaning of the peaks at 92.5, 9.5 and 5.2 ppm is discussed further in the revised manuscript (as shown below).

Revised parts in the manuscript:

In page 6, “To better understand the self-discharge behavior, as shown in Fig. S6, ^7Li SSNMR was used to probe the chemical state of $\text{Li}_{21}\text{Si}_5$ powder and $\text{Si-Li}_{21}\text{Si}_5$ powder. It is sensitive to both crystalline and amorphous phases. Based on the literature already reported by Key et al, ^[28] the homogeneous distribution of Si ions in the Li matrix gives rise to the most shielded Li environment(s), resulting in a broad resonance. The curve showed broad signals centered at 92.5, 9.5, and 5.2 ppm, the

92.5 ppm indicates the fully lithiated Si in the form of $\text{Li}_{21}\text{Si}_5$, and the 9.5 ppm and 5.2 ppm can be assigned to phases of $\text{Li}_{15}\text{Si}_4$ and $\text{Li}_{13}\text{Si}_4$, respectively. It indicates the presence of Li_xSi heterophase within the $\text{Li}_{21}\text{Si}_5$ phase. After cold-pressed sintering, the Si- $\text{Li}_{21}\text{Si}_5$ displayed a significantly enhanced signal at 9.5 ppm, suggesting that the Li-Si alloy is undergoing a transition from a high lithium state to a low lithium state.”

Reference

[28] Key. et al. Real-time NMR investigations of structural changes in silicon electrodes for lithium-ion batteries. *JACS*. **131**, 9239-9249 (2009).

6. In Page 6 and 124th line, it is difficult to conclude that the electrodes sintered at high pressure are "efficient" ionic/electronic dual conductor or have "efficient" conductive networks only through the physicochemical and morphological characterizations. It is appropriate to emphasize "efficiency" in the description part of electrochemical performance.

Response: We express thanks to reviewer for the valuable comment. We have revised the description in the revised manuscript.

Revised parts in the manuscript:

In page 6-7, “we demonstrate that the cold-pressed sintering and self-discharge effects facilitate the formation of integrated electrodes with three-dimensional ionic/electronic dual-conductor $\text{Li}_{21}\text{Si}_5$ layer and conductive networks in Si- $\text{Li}_{21}\text{Si}_5$ layer.”

7. The $\text{Li}_{21}\text{Si}_5/\text{Si-Li}_{21}\text{Si}_5$ showed the most uniform surface among the samples. Then, are there specific reasons why the roughness increases from 0.45 μm to 2.15 μm in Figure 2e-g even if more $\text{Li}_{21}\text{Si}_5$ was used in Si- $\text{Li}_{21}\text{Si}_5$ and $\text{Li}_{21}\text{Si}_5/\text{Si-Li}_{21}\text{Si}_5$ compared to pure Si.

Response: We are grateful and acknowledge the reviewer for the professional comments. The roughness increaseing is likely attributable to the markedly lower Young's modulus of $\text{Li}_{21}\text{Si}_5$ alloys in comparison to that of Si. This property renders them highly malleable, facilitating their retention on the mold surface throughout the

demolding process. In order to increase the rigor of the paper, we have explained it in the revised manuscript.

Revised parts in the manuscript:

In page 7, “The roughness increases from 0.45 μm to 2.15 μm in Fig. 2e₁-g₁, which is likely attributable to the markedly lower Young's modulus of Li₂₁Si₅ alloys in comparison to that of Si. This property renders them highly malleable, facilitating their retention on the mold surface throughout the demolding process.”

8. In Figure 3a, the overall thickness appears to be maintained at ~50 μm after lithiation. How can the thickness be maintained even though the volume expands by 300 % during lithiation?

Response: We are grateful and acknowledge the reviewer for the valuable comment. Fig. 3a shows an electrode prepared from 10 mg of Si powder with a theoretical capacity of 35.79 mAh, corresponding to a volume expansion of 300% when lithiation is complete. In this test, the lithiation capacity was only 3 mAh, which is much lower than the theoretical value, resulting in a low expansion of 15%.

9. It is recommended to measure the diffusion coefficients of Li₂₁Si₅ as a reference and the diffusion coefficients of all the anodes at room temperature.

Response: We express thanks to reviewer for the valuable comment. In Fig. 3f, we obtained the diffusion coefficient of the LCO | SSE | Li₂₁Si₅/Si-Li₂₁Si₅ cell as $2.09 \times 10^{-6} \text{ cm}^2 \text{ S}^{-1}$ (the diffusion coefficient of Li₂₁Si₅ alloy will be higher or equal to this value). Based on this, we collected the published works on the diffusion coefficients of different anodes in recent years and the results were compared as shown in Table 2. ^[1-16] It shows that the lithiated anode will have higher diffusion coefficients, and the Li₂₁Si₅ alloy has higher diffusion coefficients than them. We have added relevant results in Supplementary Information and revised them in the revised manuscript.

Supplementary Table 2 | Comparison of diffusion coefficient in this work and different anode in published literature.

References	Anode	Diffusion coefficients (cm ² s ⁻¹)	Years
1	RGR	2.0×10^{-13}	2024
2	Cr _{0.5} Nb _{24.5} O ₆₂	2.19×10^{-13}	2017
3	W ₃ Nb ₁₄ O ₄₄	8.02×10^{-13}	2019
4	Li _{0.38} Pr _{0.54} TiO ₃	1.80×10^{-12}	2024
5	NiNb ₂ O ₆	1.20×10^{-12}	2022
6	Si@Si ₃ N ₄ @C	8.11×10^{-11}	2020
7	Li ₄ Ti ₅ O ₁₂	3.89×10^{-11}	2022
8	LiCrTiO ₄	1.54×10^{-11}	2017
9	Li _{1.2} Ni _{2.5} B ₂	10^{-10} - 10^{-11}	2024
10	LiMg	1.60×10^{-10}	2022
11	Co ₂ VO ₄	3.14×10^{-10}	2022
12	MnO ₂ @PNC	10^{-8}	2022
13	Li ₂ MSiO ₄	$10^{-6.5}$ - $10^{-7.5}$	2022
14	LiC ₆	2.0×10^{-7}	2024
This work	Li₂₁Si₅	$\geq 2.09 \times 10^{-6}$	2024

Notes and references

- Chen, W.; Qu, H.; Shi, R.; Wang, J.; Ji, H.; Zhuang, Z.; Ma, J.; Tang, D.; Li, J.; Tang, J., Upcycling spent graphite into fast-charging anode materials through interface regulation. *ACS Energy Lett.* **9**, 3505-3515 (2024).
- Yang, C.; Yu, S.; Lin, C.; Lv, F.; Wu, S.; Yang, Y.; Wang, W.; Zhu, Z.-Z.; Li, J.; Wang, N., Cr_{0.5}Nb_{24.5}O₆₂ nanowires with high electronic conductivity for high-rate and long-life lithium-ion storage. *ACS nano* **11**, 4217-4224 (2017).
- Yan, L.; Shu, J.; Li, C.; Cheng, X.; Zhu, H.; Yu, H.; Zhang, C.; Zheng, Y.; Xie, Y.; Guo, Z., W₃Nb₁₄O₄₄ nanowires: ultrastable lithium storage anode materials for advanced rechargeable batteries. *Energy Storage Mater.* **16**, 535-544 (2019).
- Liu, H.; Xiao, J.; Cao, K.; Ren, N.; He, H.; Li, Y.; Si, J.; Zeng, S.; Pan, B.; Chen, C., A-site deficient perovskite lithium praseodymium titanate as a high-rate anode for lithium-ion batteries. *Chem. Eng. J.* **479**, 147765 (2024).
- Xia, R.; Zhao, K.; Kuo, L. Y.; Zhang, L.; Cunha, D. M.; Wang, Y.; Huang, S.; Zheng, J.; Boukamp, B.; Kaghazchi, P., Nickel niobate anodes for high rate lithium-ion batteries. *Adv. Energy Mater.* **12**, 2102972 (2022).
- Xiao, Z.; Lei, C.; Yu, C.; Chen, X.; Zhu, Z.; Jiang, H.; Wei, F., Si@ Si₃N₄@C composite with egg-like structure as high-performance anode material for lithium ion batteries. *Energy Storage Mater.* **24**, 565-573 (2020).
- Jin, X.; Han, Y.; Zhang, Z.; Chen, Y.; Li, J.; Yang, T.; Wang, X.; Li, W.; Han, X.; Wang, Z., Mesoporous single-crystal lithium titanate enabling fast-charging Li-Ion batteries. *Adv. Mater.* **34**, 2109356 (2022).

8. Li, X.; Huang, Y.; Li, Y.; Sun, S.; Liu, Y.; Luo, J.; Han, J.; Huang, Y., Al doping effects on LiCrTiO₄ as an anode for lithium-ion batteries. *Rsc Adv.* **7**, 4791-4797 (2017).
9. Liu, W.; Zong, K.; Ghani, U.; Saad, A.; Liu, D.; Deng, Y.; Raza, W.; Li, Y.; Hussain, A.; Ye, P., Ternary lithium nickel boride with 1D rapid-ion-diffusion channels as an anode for use in lithium-ion batteries. *Small* **20**, 2309918 (2024).
10. Siniscalchi, M.; Liu, J.; Gibson, J. S.; Turrell, S. J.; Aspinall, J.; Weatherup, R. S.; Pasta, M.; Speller, S. C.; Grovenor, C. R., On the relative importance of Li bulk diffusivity and interface morphology in determining the stripped capacity of metallic anodes in solid-state batteries. *ACS Energy Lett.* **7**, 3593-3599 (2022).
11. Ren, J.; Wang, Z.; Xu, P.; Wang, C.; Gao, F.; Zhao, D.; Liu, S.; Yang, H.; Wang, D.; Niu, C., Porous Co₂VO₄ nanodisk as a high-energy and fast-charging anode for lithium-ion batteries. *Nano-Micro Lett.* **14**, 1-14 (2022).
12. Yuan, X.; Ma, Z.; Jian, S.; Ma, H.; Lai, Y.; Deng, S.; Tian, X.; Wong, C.-P.; Xia, F.; Dong, Y., Mesoporous nitrogen-doped carbon MnO₂ multichannel nanotubes with high performance for Li-ion batteries. *Nano Energy*, **97**, 107235 (2022).
13. Zhang, J.; Yin, Q.; Wu, Y.; Zhang, S.; Wang, K.-J.; Han, J., Fe saponite, a layered silicate for reversible lithium-ions storage with large diffusion coefficient. *J. Energy Chem*, **67**, 92-100 (2022).
14. Li, X.; Ning, P.; Liu, P.; Chen, Y.; Liu, J.; Liu, W.; Wang, J.; Huang, H.; Yang, B.; Xia, X., Enabling fast mass transport in anode by a smartly built-in LiC₆ phase for high-performance solid-state lithium metal batteries. *Adv. Funct. Mater.*, 2408447 (2024).

Revised parts in the manuscript:

In page 9-10, “Table 2 is comparison of diffusion coefficient in this work and different anode in published literature. It shows that the lithiated anode will have higher diffusion coefficients, and the Li₂₁Si₅ alloy has greater diffusion coefficients than them.”

10. It is suggested to investigate the cell cycling performances at room temperature including the CCD test in symmetric cells, and rate capability and long-term cyclability in full cells.

Response: This is a constructive suggestion, and we acknowledge this comment from the reviewer. As shown in Fig. Ra and b, the Li₂₁Si₅/Si-Li₂₁Si₅-ASSB can be cycled at room temperature, but the synthetic electrolyte has a low ionic conductivity of $8.46 \times 10^{-4} \text{ S cm}^{-1}$ at room temperature, with the higher ionic conductivity of $1.87 \times 10^{-3} \text{ S cm}^{-1}$ at 45 °C (the area of the electrolyte sheet is 0.785 cm²). However,

its low ionic conductivity at 25 °C resulted in a low average CE (99.54%) of ASSB that leads to poor performance, and the capacity retention of only 80% after 45 cycles. Nonetheless, the core of our article is to highlight the $\text{Li}_{21}\text{Si}_5/\text{Si-Li}_{21}\text{Si}_5$ anode design in ASSBs. By moderately increasing the temperature, it is possible to enhance the lithium ions transport efficiency of ASSB, thereby demonstrating the advanced nature of $\text{Li}_{21}\text{Si}_5/\text{Si-Li}_{21}\text{Si}_5$ anode design.

Figure R. (a) Nyquist plots of 80 mg bilayer electrolyte at different temperatures. (b) Cycle performance of $\text{Li}_{21}\text{Si}_5/\text{Si-Li}_{21}\text{Si}_5$ -ASSBs at 25 °C.

11. In the discussion on page 11, 238th line, comparing the potential between the electrodes needs the voltage profiles of each anode.

Response: We express thanks to reviewer for the valuable comment. As shown in the Figure below, there is a widely accepted view that the voltage of the Si anode decreases with increasing lithium content. To enhance the readability of the manuscript, we have added relevant literature citations (J. Solid State Chem. 1981, 37, 271–278).

Revised parts in the manuscript:

References

29. Wen, et al. Chemical diffusion in intermediate phases in the lithium-silicon system. *J. Solid State Chem.* **37**, 271–278 (1981).

12. In Figure 5 for ASSB cycling using mold cells, how did the authors achieve free external pressure? It is helpful to provide cell design information for better understanding and pressure monitoring data.

Response: We are grateful and acknowledge the reviewer for the valuable comment. As shown in Fig. R, we have added the modeling of ASSBs at high external pressure (Fig. S1a) and at no external pressure (Fig. S1b) to the Supplementary Information.

Figure R. (a) ASSBs operating high external pressure. (b) ASSBs operating free from external pressure.

13. In Figure 5b, where does the large over potential at the beginning of 1st charging come from?

Response: We are grateful and acknowledge the reviewer for the professional

comments. We consider that the overpotential is attributable to electrochemical polarization and concentration polarization. Initially, the contact of the $\text{Li}_{21}\text{Si}_5$ layer with sulfide SSE results in the formation of low ionic conductivity products (e.g. Li_2S) at the interface, which subsequently increases the electrochemical polarization of the cell. Secondly, the $\text{Li}_{21}\text{Si}_5$ alloy on the anode contains a substantial quantity of active lithium ions, which can form a concentrated polarization with the cathode. In conclusion, these factors contribute to the generation of a considerable overpotential. To depth the logic of the manuscript, we have added the corresponding discussion in the revised manuscript.

Revised parts in the manuscript:

In page 13, “The $\text{Li}_{21}\text{Si}_5/\text{Si}-\text{Li}_{21}\text{Si}_5$ -ASSB in Fig.5b demonstrate a marked overpotential during the initial charge cycle, which is likely attributable to side reactions between the $\text{Li}_{21}\text{Si}_5$ layer and the SSE, as well as concentration polarization between the $\text{Li}_{21}\text{Si}_5$ alloy and the cathode.”

14. Is there a possible reason why the coulombic efficiency exceeds 100 % at 10.2 mA cm^{-2} in Figure S11?

Response: We express thanks to reviewer for the valuable comment. When the cathode is overloaded (60 mg cm^{-2}) in cycle operating free from external pressure, the release of lithium ions from the cathode during high-rate charging becomes more difficult. The $\text{Li}_{21}\text{Si}_5/\text{Si}-\text{Li}_{21}\text{Si}_5$ anode, on the other hand, performs much better and can promptly replenish the lithium source lost from the cathode during discharge, ultimately resulting in more than 100% CE.

15. Correct figure revision and caption are needed for Figure 5f. It is confusing to understand what the inset means and the EIS information on the anodes after cycling.

Response: We express thanks to reviewer for the valuable comment. In the EIS, the inset is a magnified view of the 0-125 ohm range, which better demonstrates the EIS information of the $\text{Li}_{21}\text{Si}_5/\text{Si}-\text{Li}_{21}\text{Si}_5$ anode. In the equivalent circuit, the impedance comparison between anode after 1 cycle (data from Fig. 3e) and 1000 cycles is further shown.

16. To claim that the anode structure and its interface remain stable in Figure 5g, it is recommended to separate the resistances of the anode and cathode.

Response: We express thanks to reviewer for the valuable comment. In the EIS test, R_{sei} and R_{ct} are the result of the coupling of multiple interfaces. As a result, it is very difficult to separate the contributions of the anode and cathode by simple mathematical analysis. In order to more accurately reflect the influence of the anode side, we selected a high-voltage stabilized halide SSE (Li_3InCl_6)^[1-2] to reduce the influence of the cathode interface. Furthermore, the cathode was composed of stabilized lithium cobaltate in a high ratio state (LCO: $\text{Li}_3\text{InCl}_6=6:4$)^[3-4] to minimize the influence of the cathode bulk phase.

Reference

1. Luo, J. et al. Rapidly in situ cross-linked poly (butylene oxide) electrolyte interface enabling halide-based all-solid-state lithium metal batteries. *ACS Energy Lett.* **9**, 3676-3684 (2023).
2. Koç, T. et al. Toward optimization of the chemical/electrochemical compatibility of halide solid electrolytes in all-solid-state batteries. *ACS Energy Lett.* **9**, 2979-2987 (2022).
3. Zhang, Z. et al. An all-electrochem-active silicon anode enabled by spontaneous Li–Si alloying for ultra-high performance solid-state batteries. *Energy Environ. Sci.* **17**, 1061–1072 (2024).
4. Yan, W. et al. Hard-carbon-stabilized Li–Si anodes for high-performance all-solid-state lithium-ion batteries. *Nat. Energy* **8**, 800–813 (2023).

17. In Figure S12, the areal capacity is only 0.1 mAh/cm² for cycling at 2C. This is barely meaningful for evaluating stability or activity.

Response: We express thanks to reviewer for the valuable comment. We have removed it from the revised manuscript.

18. Figure S14 requires an SEM image of the secondary electron mode for a clear

cross section.

Response: We are grateful and acknowledge the reviewer for the professional comments. The corresponding secondary electron SEM image is shown in Fig. 6e. In order to enhance the readability of the manuscript, we have added a note in the **Supplementary Information**.

19. Items that need correction.

- 1) Page 2, 61st line: lithium-ions → lithium ion
- 2) Page 6, 140th line: a relatively stable a stable sulfide
- 3) Page 7, 148th line: The areal capacity should be expressed as “mAh cm⁻²”.
- 4) Page 8, 168th line: charge-transfer resistance at SEI (RSEI) → ion-transfer resistance at SEI (RSEI)
- 5) Page 9, 187th line: a “slow” increase in polarization → The authors seem to want to express “rapid” increase.
- 6) Page 9, 195th line: micrometer → micrometer-sized
- 7) Page 9, 197th line: TEM → The first notation of the abbreviation requires the full name of TEM.
- 8) Page 11, 232nd line: HRTEM → The first notation of the abbreviation requires the full name of HRTEM.
- 9) Figure 4h: The labeling of sample name needs to be changed from “Li₂₁Si₅/Si-Li₂₁Si₅ after cycling” to “Li₂₁Si₅ after cycling”.

Response: Thanks for your careful and constructive suggestion. We have corrected them in the revised manuscript and carefully checked it in full. Moreover, the language of the entire manuscript has been polished to meet the high-quality of this journal.

Reviewer #4 (Remarks to the Author):

I co-reviewed this manuscript with one of the reviewers who provided the listed reports. This is part of the Nature Communications initiative to facilitate training in peer review and to provide appropriate recognition for Early Career Researchers who

co-review manuscripts.

Response: We are grateful and acknowledge the reviewer for the professional comments. The point-by-point responses to the comments are provided above, and the manuscript has been revised accordingly.

Responses to Reviewers' comments

Reviewer #1:

The authors have conducted additional experiments to address my concerns. Unfortunately, it remains unclear what advantages this study offers. Most importantly, there is no direct evidence to support the authors' key claim that introducing an additional Li₂₁Si₅ interlayer enables operation without external pressure. I cannot support the publication of this paper until the following issues are resolved:

Response: Thank you very much for your efforts in reviewing the manuscript and for your professional comments. We have made a point-to-point response and revised our manuscript accordingly.

1. The cell performances at different N/P ratios were shown in the revised paper. However, even at an N/P of 3.3, low reversible capacity and initial efficiency were observed, and the cycle performance at an N/P of 1.6 was poor. I think that the good performance at high N/P is due to the available Li source in the anode electrode. To stably operate at an areal capacity of 2.8 mAh/cm², it can be estimated that an anode thickness of over 80 μm seems to be required. Please present the achievable energy density through this cell design.

Response: We express thanks to the reviewer for the valuable comments.

1) To the best of our knowledge, our results, even at a N/P of 3.3, represent **the highest performance reported thus so far for external pressure-free Si-ASSBs.**

Figure R1. Cycle performance of Li₂₁Si₅/Si-Li₂₁Si₅-ASSBs with a N/P ratio of 3.3.

To further demonstrate the advancement of its performance at an N/P of 3.3, Figure R1 further displays its subsequent cycling performance. The ICE was 91.72% and the capacity retention was 80% after 171 cycles. Compared with the cycling performance of N/P of 6.7, its ICE and capacity retention only decreases by 5.1% and 6.5%, respectively, indicating that cycle performance of the ASSBs can be significantly enhanced by controlling the degree of lithiation of Si (When the lithiation of Si in the

anode is about 1/3, the corresponding N/P ratio is 3.3). Nonetheless, as shown in Figure R2 and Table R1, its ICE and capacity retention remains at the leading edge of pressure-free Si-ASSBs [30-40, 22, 41].

Figure R2. Comparison of the external pressure, ICE, and cycle number between ASSBs at different N/P and published literatures. Noted that the cycle number for 80% capacity retention. The comparative literatures are based on Figure 5i in the revised manuscript.

Table R1 | Comparison of key parameters Figure R2.

Ref.	Stack pressure (MPa)	Cycle number	ICE	Anode	Cathode
30	20	100	51%	Si	NCM-111
31	2	35	92%	Li	NMC
32	70	100	96.2%	Si	NCA
33	20	50	84%	Si	NCM
34	75	100	88.5%	Li-In	NCM
35	20	45	88.7%	Si	LCO
36	2	30	86%	MXene/Mg	NCM-811
37	5	100	69.5%	Li	NCA
38	0	80	60%	Si	LFP
39	50	500	77.92%	Si	NCM-811
40	50	62	74.8%	Si	NCM
22	50	600	97.8%	Si	LCO
41	15	60	38.7%	Si	NCM
This work	0	171	91.72%	Si	LCO

2) We have added experiments to demonstrate that the good performance at high N/P is **due to the structural design advantages** of the $\text{Li}_{21}\text{Si}_5/\text{Si-Li}_{21}\text{Si}_5$ anode, not

due to the available Li source in the anode.

Figure R3. Charge-discharge profiles of ASSBs at 2.3 mA cm⁻² with Si-Li₂₁Si₅ anode (Li₂₁Si₅ mass ratio of 75%) (a), pure Li₂₁Si₅ anode (b), and Li metal anode (c).

As shown in Figure R3, 20 mg of Si-Li₂₁Si₅ powder (Li₂₁Si₅ mass ratio of 75%), 20 mg of pure Li₂₁Si₅ powder, and Li metal were used as anodes for ASSBs operating free from external pressure and their charge-discharge performance was tested at 0.1 C. With the increase of lithium content in the three anodes, the charging plateau of the ASSBs gradually increases from 3.8 V to 3.9 V, and the discharging plateau shifts from 4.2-3.2 V to 4.2-3.7 V. This suggests that the active Si in the Si-Li₂₁Si₅ anode participates in the whole process of the alloying and dealloying reactions. However, due to the lack of suitable structural design of the above anodes, the bulk phase structure of Si-Li₂₁Si₅ anode and the interfaces of pure Li₂₁Si₅ and Li metal anode would be destroyed during the lithiation process, making their ICEs 73.4%, 82.99%, and 89.27%, respectively, which are much lower than that of Li₂₁Si₅/Si-Li₂₁Si₅ anode at 97.69%. Nonetheless, all the above three ASSBs suffered soft short circuits in the subsequent cycles.

The above experiments once again verified the importance of structural design and the participation of Si in the Li₂₁Si₅/Si-Li₂₁Si₅ anode.

3) Energy density calculation of the Li₂₁Si₅/Si-Li₂₁Si₅-ASSBs.

Figure R4. Cross sectional view of the $\text{Li}_{21}\text{Si}_5/\text{Si-Li}_{21}\text{Si}_5$ -ASSB.

The energy density is calculated based on the above ASSB configuration (Figure R4), which contains the composite cathodes ($\text{Li}_3\text{InCl}_6/\text{LCO}$ or others), the SSE ($\text{Li}_3\text{InCl}_6/\text{Li}_6\text{PS}_5\text{Cl}$ or others), and the $\text{Li}_{21}\text{Si}_5/\text{Si-Li}_{21}\text{Si}_5$ anode. In general, the energy density of a ASSB is the energy density of composite cathode divided by the total mass. The SSB energy density can be described as

$$E_{ASSE} = \frac{C_c \times V_c}{m_a + m_s + m_c}$$

in which C_c is the capacity of the composite cathode; V_c is the average working voltage of the composite cathodes (3.7 V for Si-based anode | SSE | LCO); m_a , m_s , and m_c are the total weight of the anode (10-20 mg cm^{-2}), SSE, and cathode, respectively. The calculation details are shown in Table R2,

Table R2 | The energy density calculations of the $\text{Li}_{21}\text{Si}_5/\text{Si-Li}_{21}\text{Si}_5$ -ASSB.

	Weight	Weight	Weight	Weight
Cathode	$m_{c1}=15\sim45$ mg (LCO)	$m_{c1}=15\sim30$ mg (NCM)	$m_{c1}=15\sim30$ mg (NCM)	$m_{c1}=15\sim30$ mg (NCM)
	$m_{c2}=10\sim30$ mg (Li_3InCl_6)	$m_{c2}=1.5\sim3.5$ mg (SSE+Sp)	$m_{c2}=1.5\sim3.5$ mg (SSE+Sp)	$m_{c2}=1.5\sim3.5$ mg (SSE+Sp)
SSE	$m_s=80$ mg	$m_s=30$ mg	$m_s=20$ mg	$m_s=10$ mg
Anode	20 mg (N/P = 6.7)	10 mg (N/P = 3.3)	10 mg (N/P = 3.3)	10 mg (N/P = 3.3)
Total weight	125~175 mg	56.5~73.5 mg	46.5~63.5 mg	36.5~53.5 mg
Voltage	3.7 V	3.6 V	3.6 V	3.6 V
Capacity	2.2~6.1 mAh	3~6 mAh	3~6 mAh	3~6 mAh
Achieved energy density	65~128 Wh kg^{-1}			
Calculated energy density		191.2~293.9 Wh kg^{-1}	232.3~340.2 Wh kg^{-1}	295.9~403.7 Wh kg^{-1}

The cycling of ASSBs operating free from external pressure presents a significant

challenge for practical application, and in this manuscript, we focus on the structure design of the anode. In order to minimize the impact of the SSE and cathode on cycling performance, a double-layer SSE and a lower energy density cathode (LCO) were employed due to their relatively more stable performance. It is inevitable that such a strategy will result in a ASSB with a lower energy density (as shown in Table R2).

Nevertheless, it is our contention that the integration of thin SSE and high-specific-capacity cathodes (e.g. NCM) through dry electrode process and coating strategies can serve to enhance the energy density of ASSBs. ^[1-3] We would like to express our gratitude to the reviewers for their constructive comments, which will encourage our future research and development in this area.

Notes and references

- [1] K Liu, et al. A 3 μm -ultrathin hybrid electrolyte membrane with integrative architecture for all-solid-state lithium metal batteries. *Adv. Energy Mater.* **14**, 2303940 (2024).
- [2] L Hu, et al. Fusion bonding technique for solvent-free fabrication of all-solid-state battery with ultrathin sulfide electrolyte. *Adv. Mater.* **36**, 2401909 (2024).
- [3] Tan, D. H. S. et al. Carbon-free high-loading silicon anodes enabled by sulfide solid electrolytes. *Science* **373**, 1494–1499 (2021).

Revised parts in the manuscript:

In page 13, “As shown in Fig. S13a-c, the same phenomenon was observed in higher lithium content anode.”

Supplementary Figure 13. Charge-discharge profiles of ASSBs at 2.3 mA cm^{-2} with Si- $\text{Li}_{21}\text{Si}_5$ anode ($\text{Li}_{21}\text{Si}_5$ mass ratio of 75%) (a), pure $\text{Li}_{21}\text{Si}_5$ anode (b), and Li metal anode (c).

2. There is no data demonstrating that good cycle performance was achieved without external pressure. The authors should provide real-time pressure monitoring data at a high reversible capacity of 3.0 mAh/cm^2 .

Response: We express thanks to the reviewer for the valuable comment. We have added a stress test for Si- $\text{Li}_{21}\text{Si}_5$ -ASSB and $\text{Li}_{21}\text{Si}_5/\text{Si-Li}_{21}\text{Si}_5$ -ASSB with real-time stress monitoring equipment. As shown in Figure Ra, after applying the minimum pressure load of the equipment to Si- $\text{Li}_{21}\text{Si}_5$ -ASSB, the initial pressure of Si- $\text{Li}_{21}\text{Si}_5$ -ASSB was 0.6 MPa, and it failed work after one cycle. During the first charging process, due to the lithiation expansion of Si, the pressure increased to 0.96 MPa, corresponding to an increment of 60%. After the discharge process completed, the pressure decreases to 0.67 MPa. In contrast, as shown in Figure Rb, the initial pressure of $\text{Li}_{21}\text{Si}_5/\text{Si-Li}_{21}\text{Si}_5$ -ASSB is 0.54 MPa, and after the first charge, the pressure increases to 0.65 MPa, corresponding to an increment of 20%. After discharge, the pressure decreased to 0.52 MPa. In the second cycle, the rate of pressure change is only 9.6%. The above phenomenon indicates that the $\text{Li}_{21}\text{Si}_5$ layer can reduce the cyclic expansion stress of the ASSBs, which is beneficial for the stability of its performance.

Figure R. Real-time pressure monitoring for Si-Li₂₁Si₅-ASSB (a) and Li₂₁Si₅/Si-Li₂₁Si₅-ASSB (b).

Revised parts in the manuscript:

In page 13-15, “The real-time pressure monitoring data of Si-Li₂₁Si₅-ASSB and Li₂₁Si₅/Si-Li₂₁Si₅-ASSB was shown in Fig. S14a-b, respectively. After applying the minimum pressure load of the equipment to Si-Li₂₁Si₅-ASSB, the initial pressure of Si-Li₂₁Si₅-ASSB was 0.6 MPa, and it failed work after one cycle. During the first charging process, due to the lithiation expansion of Si, the pressure increased to 0.96 MPa, corresponding to an increasement of 60%. After the discharge process completed, the pressure decreases to 0.67 MPa. In contrast, as shown in Fig. S14b, the initial pressure of Li₂₁Si₅/Si-Li₂₁Si₅-ASSB is 0.54 MPa, and after the first charge, the pressure increases to 0.65 MPa, corresponding to an increasement of 20%. After discharge, the pressure decreased to 0.52 MPa. In the second cycle, the rate of pressure change is only 9.6%. The above phenomenon indicates that the Li₂₁Si₅ layer can reduce the cyclic expansion stress of the ASSBs, which is beneficial for the stability of its performance.”

Supplementary Figure 14. Real-time pressure monitoring for Si-Li₂₁Si₅-ASSB (a) and Li₂₁Si₅/Si-Li₂₁Si₅-ASSB (b).

3. The expansion/contraction of Si during cycling is inevitable. Even if the additional Li₂₁Si₅ layer helps with uniform Li⁺ flux at the interface between the electrolyte layer and the anode electrode, the increase in interfacial resistance between silicon and the Li₂₁Si₅ matrix within the anode electrode could not be avoided. How to solve the expansion/contraction of Si during cycling?

Response: Thank you for your professional comments. In the Si anode of a liquid lithium-ion battery, the Si particles are lithiated and de-lithiated with cyclic volume expansion/contraction. At this stage, the Si anode is still composed of numerous active Si particles, which are constantly expanding and fragmenting.

In contrast, the electrochemical behavior of the Si particles is distinctly different in Si-based ASSBs. As previously reported,^[1-3] it has been noted that the lithiation of a pure Si anode under high external pressure is accompanied by **electrochemical sintering**. In this case, under the combined effect of high expansion and high external pressure, neighboring Li_xSi particles will squeeze with each other. As the degree of lithiation increases, the Young's modulus of Li_xSi particles will further decrease, which in turn accelerates the fusion between the particles and ultimately transforms into an integrated amorphous Si film anode. Moreover, the low Young's modulus of Li₂₁Si₅ alloy can provide a stress buffer for volume expansion of Si, and the self-discharge effect can create volume exchange, both of which are beneficial for stress release.

Figure R. Cross-section SEM images of Si (a), Si-Li₂₁Si₅ (b), and Li₂₁Si₅/Si-Li₂₁Si₅ (c) anode after lithiation.

As shown in Figure Ra, without ion/electronic conductive network, the pure Si anode will exhibit a top-down plating behavior, and the expansion stress will be concentrated at the lithiation interface, revealing that an external pressure is needed to facilitate uniform electrochemical sintering. In this work, to cycle Si anode without external pressure, we first added a low Young's modulus Li₂₁Si₅ network into the bulk part of the anode to construct an efficient ion/electronic conductive network, and the lithium ions can be rapidly transported within the Si anode during the subsequent cycling process, thus realizing an integrated electroplating behavior (Figure Rb). However, this reaction produces a localized stress concentration during the electrochemical sintering process, leading to the formation of local cracks inside the electrode. Furthermore, through the incorporation of a high ionic/electronic conductive Li₂₁Si₅ layer on the anode surface, the expansion stress during the electrochemical sintering process was homogenized (Figure Rc). It is concluded that, with the unique design of Li₂₁Si₅ interlayer and incorporation of Li₂₁Si₅ particles, the electrochemical sintering is homogenized thus release the stress during lithiation/delithiation.

Notes and references

- [1] Zhang, Z. et al. An all-electrochem-active silicon anode enabled by spontaneous Li-Si alloying for ultra-high performance solid-state batteries. *Energy Environ. Sci.* **17**, 1061–1072 (2024).
- [2] W. Yan, et al. Hard-carbon-stabilized Li-Si anodes for high-performance all-solid-state Li-ion batteries. *Nat. Energy* **8**, 1-14 (2023).
- [3] Tan, D. H. S. et al. Carbon-free high-loading silicon anodes enabled by sulfide solid electrolytes. *Science* **373**, 1494–1499 (2021).

4. In the authors' previous paper (EES, 2024, 17, 1061–1072), the areal capacity of 18 mAh/cm² with the high CE over 90 % was achieved even without a Li₂₁Si₅ protective layer. Then, with the addition of the Li₂₁Si₅ layer, is operation without external

pressure possible at high capacities above 10 mAh/cm²?

Response: We express thanks to the reviewer for the valuable comment. When the battery is cycled without external pressure, the expansion of the Si anode and the interface issues are the most serious problems. In this work, those problems on the anode were mostly solved by structural design and the addition of the Li₂₁Si₅ layer, so that a high capacity of 10 mAh cm⁻² was demonstrated in the cycling of a symmetric cell.

Figure R. (a) EIS tests before cycling for Li₂₁Si₅/Si-Li₂₁Si₅-ASSBs with cathode mass loadings of 30 mg cm⁻², 45 mg cm⁻², 60 mg cm⁻², and 90 mg cm⁻², respectively. Charge-discharge profiles at 2.5 mA cm⁻² for cathode mass loadings of 45 mg cm⁻² (b), 60 mg cm⁻² (c), and 90 mg cm⁻² (d), respectively. All of the above tests were conducted at 45 °C.

However, as the mass loading in the cathode increases, the thickness also increases, making the original ionic/electronic conductive system at the cathode cannot work efficiently without external pressure, which can lead to over-polarization of the battery and inability to utilize the full capacity. Figure Ra illustrate the EIS tests before cycling for Li₂₁Si₅/Si-Li₂₁Si₅-ASSBs with cathode mass loadings of 30 mg cm⁻², 45 mg cm⁻², 60 mg cm⁻², and 90 mg cm⁻², respectively. Their impedances were 70.2 Ω, 162.5 Ω, 364.3 Ω, and 614.7 Ω, respectively, and it could be clearly noticed that the impedance of the cathode increases sharply at high mass loadings. As shown in Figure Rb-d, although the mass loading of the cathodes has increased, both ICE and cycling stability have decreased. Not only that, the increase in polarization further

reduces their reversible capacity, indicating that the performance of the cell with high mass loading is limited by the ionic/electronic transport efficiency of the cathode.

In the follow-up work on Si-based ASSBs operation without external pressure, we encourage the development of suitable electrolytes and cathodes using the cold-pressure sintering idea to further optimize the ionic/electronic conductive system and enhance the performance of the battery. We thank the reviewers again for their in-depth insights and inspirations.

5. When the volume ratio of $\text{Li}_{21}\text{Si}_5$ in the anode composite increases, can the same effect be expected as using an additional $\text{Li}_{21}\text{Si}_5$ interlayer.

Response: We express thanks to reviewer for the valuable comment. As shown in Figure R, 20 mg of Si- $\text{Li}_{21}\text{Si}_5$ powder ($\text{Li}_{21}\text{Si}_5$ mass ratio was 75%, higher than the 50% in the manuscript) was used as anodes for ASSBs operating free from external pressure and the charge-discharge performance was tested at 0.1 C. However, due to the lack of suitable structural design, its ICE was only 73.4%, which were much lower than that of $\text{Li}_{21}\text{Si}_5/\text{Si-Li}_{21}\text{Si}_5$ anode at 97.69%. Not only that, the ASSBs suffered soft short circuits in the subsequent cycles. It once again verified the importance of structural design and the participation of Si in the $\text{Li}_{21}\text{Si}_5/\text{Si-Li}_{21}\text{Si}_5$ anode.

Figure R. Charge-discharge cycle curves of ASSBs with Si- $\text{Li}_{21}\text{Si}_5$ anode ($\text{Li}_{21}\text{Si}_5$ mass ratio of 75%).

Reviewer #3:

The authors sought to address most of the issues related to the in-depth understanding of the $\text{Li}_{21}\text{Si}_5/\text{Si}-\text{Li}_{21}\text{Si}_5$ double-layered anode design. For reconsideration, we encourage the authors to address the following concerns further:

Response: Thank you for the positive recommendation on our paper. We acknowledge reviewer for their constructive comments. The manuscript has been revised accordingly with point-by-point responses to the comments.

1. What pressure was used for the self-discharge test in Supplementary Figure 5? Sintering of $\text{Si}/\text{Li}_{21}\text{Si}_5$ in a lithium metal half cell under high pressure can cause lithium metal to creep throughout the SSE layer, leading to a short circuit. Please provide the test conditions in Figure or Experimental to avoid confusion.

Response: We express thanks to reviewer for the valuable comment. During the self-discharge test in Supplementary Figure 5, cold-pressed sintering of the anode was done at high external pressure, and the self-discharge test was performed without external pressure, and lithium metal does not creep. We have added instructions in the supplementary material.

Revised parts in the manuscript:

In page 6, “As shown in Fig. S5, when $\text{Si}-\text{Li}_{21}\text{Si}_5$ is cold-sintered for a long time and then assembled into a ASSB with lithium metal, lithium ions in the $\text{Li}_{21}\text{Si}_5$ alloy continue to enter the bulk phase of Si to form Li_xSi .”

2. It is suggested to switch the numbers between the two supplementary tables.

Response: Thanks for your careful and constructive suggestion. We have switched the numbers between the two supplementary tables in the revised manuscript.

Responses to Reviewers' comments

We express thanks to editor and reviewer for the valuable comment. In response to your valuable suggestion that “please be sure to include the pressure monitoring data for the cells operating under zero pressure”, we would like to explain the following.

During our attempts to test the real-time pressure monitoring of ASSB, we found that the monitoring data was consistently 0 MPa without any limit on the thickness of the ASSB. Therefore, when attempting to ascertain the pressure monitoring data of the ASSB, it is necessary to limit the thickness of ASSB, which in turn improves the accuracy of the test data. Based on this, we applied a pressure of less than 1 MPa to it. In fact, the internal pressure of a commercial lithium-ion battery can be approximated to 1 MPa after evacuation and encapsulation.

Reviewer #1:

The authors have tried to address my concerns. Compared to the previous study, they claim that the use of the $\text{Li}_{4.4}\text{Si}$ layer enables stable cycling without external pressure. However, the current Figure S14 does not adequately demonstrate stable cell operation under zero pressure. The authors should provide pressure monitoring data with charge-discharge profiles under stable cell operation conditions. If they can provide pressure monitoring data for 5 cycles at least, I would support publication of this paper.

Response: We express thanks to the reviewer for the valuable comment. As shown in Figure R1, limited by the testing instrument, the pressure testing data we provided before was tested at room temperature with ~ 60% air humidity (Figure R1a). At this point, in a mold with no external pressure that poor sealing, moisture and oxygen in the atmosphere are highly susceptible to reacting with the solid electrolyte, as evidenced by the huge resistance increase after 15h (Figure R1b), which in turn affects the cycle stability of the $\text{Li}_{21}\text{Si}_5/\text{Si}-\text{Li}_{21}\text{Si}_5$ -ASSB (Figure R1c).

Figure R1. (a) Instruments used for real-time pressure monitoring in constant thickness mode. Testing conditions: 20 °C, air atmosphere, air humidity of 60%. (b) EIS tests of SSE in the above environment. (c) Real-time pressure monitoring for $\text{Li}_{21}\text{Si}_5/\text{Si}-\text{Li}_{21}\text{Si}_5\text{-ASSB}$.

In order to further minimize the impact of the environment on the $\text{Li}_{21}\text{Si}_5/\text{Si}-\text{Li}_{21}\text{Si}_5\text{-ASSB}$, we used another hand-made set of pressure-sensitive sensor (Figure R2a), which could be placed in a thermostat box to remain stable temperature at 45 °C and low air humidity below 10%. At this condition, the side reactions of the solid electrolyte significantly reduced and the cycling stability of the battery achieved, as confirmed by the relative stable resistance during a long time of 50 h in Figure R2b. The real-time pressure monitoring data and stable cycling performance were shown in Figure R2c. After applying the minimum pressure (0.8 MPa) load of the equipment to limit the thickness of $\text{Li}_{21}\text{Si}_5/\text{Si}-\text{Li}_{21}\text{Si}_5\text{-ASSB}$, the pressure increased by 1.51 MPa in the first charge. In the next cycles, the rate of pressure change was slowly decreasing.

Figure R2. (a) Instruments used for real-time pressure monitoring in pressure-sensitive sensing mode. Testing conditions: 45 °C, air atmosphere, air humidity: <math><10\%</math>. (b) EIS tests of SSE in the above environment. (c) Real-time pressure monitoring for $\text{Li}_{21}\text{Si}_5/\text{Si-Li}_{21}\text{Si}_5\text{-ASSB}$.

Revised parts in the manuscript:

In page 13-15, “The real-time pressure monitoring data of $\text{Li}_{21}\text{Si}_5/\text{Si-Li}_{21}\text{Si}_5\text{-ASSB}$ was shown in Fig. S14. After applying the minimum pressure load (0.8 MPa) of the equipment to limit the thickness of $\text{Li}_{21}\text{Si}_5/\text{Si-Li}_{21}\text{Si}_5\text{-ASSB}$, the pressure increased by 1.51 MPa in the first charge. In the next cycles, the rate of pressure change was slowly decreasing.”

Supplementary Figure 14. Real-time pressure monitoring for Li₂₁Si₅/Si-Li₂₁Si₅-ASSB. Testing conditions: 45 °C, air atmosphere, air humidity: <10%.